# TIPO: TEXT TO IMAGE WITH TEXT PRESAMPLING FOR PROMPT OPTIMIZATION

**Shih-Ying Yeh**[⋆‡♠♦]   **Yi Li**[⋆♣]   **Sang-Hyun Park**[♡]   **Giyeong Oh**[♡]   **Xuehai Wang**[⋆]

**Min Song**[♡◇]          **Youngjae Yu**[†]          **Shang-Hong Lai**[♠]

National Tsing Hua University[♠]  Nanyang Technological University[♣]  Yonsei University[♡]
Onoma AI[◇]  Karolinska Institute[⋆]  Seoul National University[†]  Comfy Org Research[♦]
Corresponding Author[‡]  Equal Contribution[⋆]

kohaku@kblueleaf.net • Code: github.com/KohakuBlueleaf/KGen

## ABSTRACT

TIPO (Text-to-Image Prompt Optimization) introduces an efficient approach for automatic prompt refinement in text-to-image (T2I) generation. Starting from simple user prompts, TIPO leverages a lightweight pre-trained model to expand these prompts into richer, detailed versions. Conceptually, TIPO samples refined prompts from a targeted sub-distribution within the broader semantic space, preserving the original intent while significantly improving visual quality, coherence, and detail. Unlike resource-intensive methods based on large language models (LLMs) or reinforcement learning (RL), TIPO provides computational efficiency and scalability, opening new possibilities for effective, automated prompt engineering in T2I tasks. Extensive experiments across multiple domains demonstrate that TIPO delivers stronger text alignment, reduced visual artifacts, and consistently higher human preference rates, while maintaining competitive aesthetic quality. These results highlight the effectiveness of distribution-aligned prompt engineering and point toward broader opportunities for scalable, automated refinement in text-to-image generation.

## 1 INTRODUCTION

The rapid proliferation of Text-to-Image (T2I) generative models has revolutionized artistic creation (Ossa et al., 2024; Betker et al., 2023; Esser et al., 2024a; Saharia et al., 2022; Ramesh et al., 2021; 2022; Shi et al., 2020; Rombach et al., 2022a; Podell et al., 2024; Sauer et al., 2024; Chen et al., 2024b;a; Li et al., 2024b; Esser et al., 2024b; black-forest labs, 2024). These models offer direct control over generative visual content via *text prompts*. To achieve precise control, modern T2I architectures are often trained on lengthy, detailed text descriptions, which may consist of individual, formatted tags of objects, backgrounds, styles, or complex, integrated paragraphs outlining image content and layout. However, the increasing complexity of prompts often forces users to iteratively refine them to convey intent. Moreover, most state-of-the-art T2I models are aesthetically fine-tuned (e.g., on LAION-aesthetics (Podell et al., 2024)) to favor nuanced artistic and stylistic cues, making high-quality T2I artwork mainly accessible to those with significant artistic expertise.

Extensive efforts have been made to reduce the reliance on human expertise through *prompt optimization*, i.e., expanding and refining a user's primitive input into a more detailed prompt to enhance generation quality. As shown in Figure 1(a), a straightforward approach is to leverage pre-trained Large Language Models (LLMs) to rewrite prompts in a zero-shot manner (Mañas et al., 2024). Yet, LLMs are primarily trained on general natural language, such as paragraphs and dialogues, which differ significantly from the structured prompts used for T2I models. This discrepancy often leads to additional effort in crafting LLM prompts and increased misalignment between generated images and intended prompts. Figure 1(b) shows a more effective approach: training LLMs directly on prompt data collected from model users (AUTOMATIC, 2022; daspartho, 2022). While promising, this method is inherently constrained by the varying levels of user expertise, often resulting in inconsistent or suboptimal outputs. Recent work (Hao et al., 2023) trains LLMs with reinforcement learning, where aesthetic scores of generated images serve as rewards, as depicted in Figure 1(c).

However, reinforcement learning is performed on one specific T2I model with high computational cost, hindering its application to a broader variety of models.

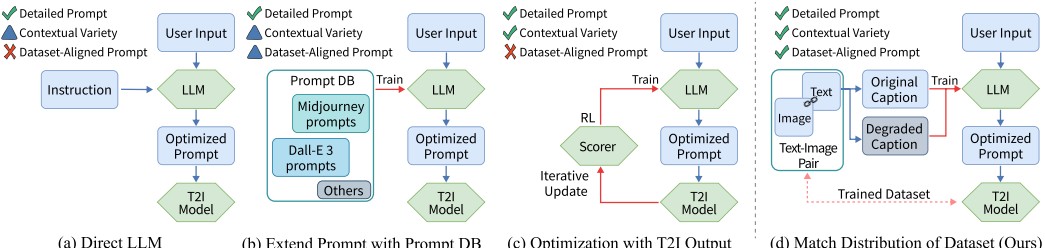

Figure 1: Comparison of prompt optimization methods using LLM. **(a)** uses instructions for prompting but its understanding is constrained by the LLM's knowledge base, not the T2I model. **(b)** relies on a curated prompt database, enhancing detail but limiting variety by not fully leveraging the T2I model's learned distribution. **(c)** optimizes using the scorer with RL, requiring multi-turn inference with additional cost. **(d)** aligns prompts with the T2I model's training distribution, ensuring detailed and diverse prompt generation that fits the target T2I model.

In contrast to prior work, we argue that *good prompts should align with the large-scale text distributions of T2I models' training*, including those emphasized in aesthetic fine-tuning. Such alignment allows models to better interpret user intent and leverage their learned priors, resulting in stronger text alignment and overall image quality. Based on this insight, we introduce **TIPO** (**T**ext to **I**mage with text pre-sampling for **P**rompt **O**ptimization), a framework that brings prompt optimization into the domain of large-scale multi-task pre-training. TIPO is supported by a curated 30M-pair, 40B-token caption corpus, which is filtered and balanced to maximize compatibility with leading T2I models and to preserve aesthetic quality. On top of this corpus, we design a suite of pretext tasks that reformulate raw user inputs, including both concise natural sentences and tag-based prompts, into enriched and distribution-consistent forms. Through this multi-task sampling pipeline, a lightweight language model expands (rather than fully rewrites) user prompts, preserving original semantics while enriching details that are diverse, edit-friendly, and aligned with T2I training distributions. Extensive experiments on both in-domain and out-of-domain prompts show that TIPO consistently outperforms strong baselines, achieving a 62.8% win rate in human preference (validated by more than 1,400 pairwise comparisons from 221 volunteers), and providing up to a 29.4% runtime efficiency improvement. Figure 1 illustrates the conceptual differences between existing methods and TIPO. To summarize, our contributions are at least threefold:

1. We introduce TIPO, a prompt optimization framework that leverages the large-scale text distributions used in text-to-image (T2I) training.

2. We train a lightweight multi-task language model that progressively refines both tag-based and natural language inputs into unified prompts, enhancing compatibility across a broad spectrum of T2I models.

3. Extensive experiments demonstrate that TIPO achieves superior image quality, stronger text alignment, higher human preference, competitive aesthetic quality, and improved runtime efficiency against strong baselines with SOTA T2I models, highlighting its practical value.

## 2 RELATED WORK

Prompt optimization for T2I models typically leverages language models, which can be broadly classified into two categories: (1) *Model-specific strategies* that tailor prompts for a particular T2I model, and (2) *Universal strategies* that improve prompt quality across a variety of T2I models.

**Model-specific Strategies** T2I models generate images whose quality is often measured using metrics such as fidelity, aesthetics, and user preference. These metrics facilitate reinforcement learning approaches that optimize prompts for a specific T2I model. For instance, Promptist (Hao et al., 2023) fine-tunes a pre-trained language model by using CLIP relevance scores as rewards. Similarly, PAE (Mo et al., 2024) extends this approach by generating dense text embeddings rather than discrete

text tokens, with additional control vectors during online reinforcement learning. However, these methods are computationally intensive, often struggling with a larger number of training prompts. Moreover, a model optimized for one specific T2I system may not generalize well to others. In contrast, our method leverages over 30 million text descriptions to cover a wide range of high-quality prompts, ensuring compatibility with a broad spectrum of T2I models.

**Universal Strategies**    To reduce the dependency on specific T2I models, some researchers have focused on refining prompts solely using language models. For example, CogView3 (Zheng et al., 2024) employ GLM-4 (GLM et al., 2024), and Lee et al. (2024) employ GPT-J and Text Style Transfer (TST) techniques, respectively, for prompt enhancement. However, both of them rely heavily on the LLM's inherent understanding of visual content descriptions, which may result in a misalignment with the diverse requirements of various T2I models. Alternatively, other approaches collect high-quality prompts from T2I model users to fine-tune or train LLMs (AUTOMATIC, 2022; succintly, 2022; daspartho, 2022). Such methods, however, are limited by the inconsistent expertise of users. More recently, He et al. (2025); Liu et al. (2024b) leverage vision language models to optimize prompts in an iterative loop: a user prompt generates images via a T2I model, the prompt–image pairs are evaluated by the VLM to suggest refinements, and the revised prompts are reapplied to T2I generation for continual improvement. While such iterative refinement can improve quality, the distribution mismatch between VLMs and T2I training data persists. Conversely, our approach constructs both tag-based and natural language prompts using a large-scale dataset of image-text descriptions, thereby aligning closely with the text distributions underlying T2I models.

## 3 PRELIMINARIES

We present the formal definition of T2I models and the problem statement of this work.

**Text-to-Image Model**. A *text-to-image (T2I) model* defines a conditional distribution

$$\mathcal{I}_p = P(x \mid p), \quad x \in \mathcal{X},$$

which maps a prompt $p$ to a distribution of images over the space of all possible images $\mathcal{X}$.

**Problem statement**. Let $\mathcal{I}_u$ denote the user's *intended distribution* over $\mathcal{X}$. The task of *prompt optimization* is to find an optimized prompt $p_o$ from the space of all possible prompts $\mathcal{P}$ to minimize a distance $d$ between the T2I output distribution $\mathcal{I}_p$ and the intended distribution $\mathcal{I}_u$:

$$p_o \ = \ \arg\min_{p \in \mathcal{P}} \ d\big(\mathcal{I}_p, \mathcal{I}_u\big),$$

where $d(\cdot, \cdot)$ is a distance between image distributions (e.g., Fréchet Inception Distance).

## 4 METHODOLOGY

We aim to optimize user prompts to enhance image generation quality. Instead of end-to-end optimizations tailored to a single T2I model, our focus is on prompt rewriting that generalizes across a broad spectrum of models. Our core intuition is that an ideal prompt should align with the texts used in T2I model training. However, rapid advances in image captioning have rendered these texts increasingly diverse and complex. To address this, we propose to (1) design a clearly structured prompt schema compatible with most text descriptions, and (2) implement a pre-sampling algorithm that progressively refines arbitrary, coarse user input into organized, fine-grained prompts.

### 4.1 TEXT SET PREPARATION

Although the text descriptions used for T2I model training are notably diverse, most are image captions that fall into two broad categories: tag-based and natural language (NL)-based captions. Tag-based captions, such as those in the Danbooru2023 dataset (nyanko202, 2023; Yeh, 2024a), use comma-separated, succinct terms to describe image content. In contrast, NL-based captions, typically generated by language models with visual capabilities (Liu et al., 2023; Agrawal et al., 2024; OpenAI, 2024; Li et al., 2024a; Bai et al., 2023; Dai et al., 2024; Xiao et al., 2024; Deitke et al., 2024), may comprise multiple sentences. We represent both types using a unified text set $T = \{t_1, t_2, t_3, \ldots, t_n\}$, where each element is an individual tag or sentence.

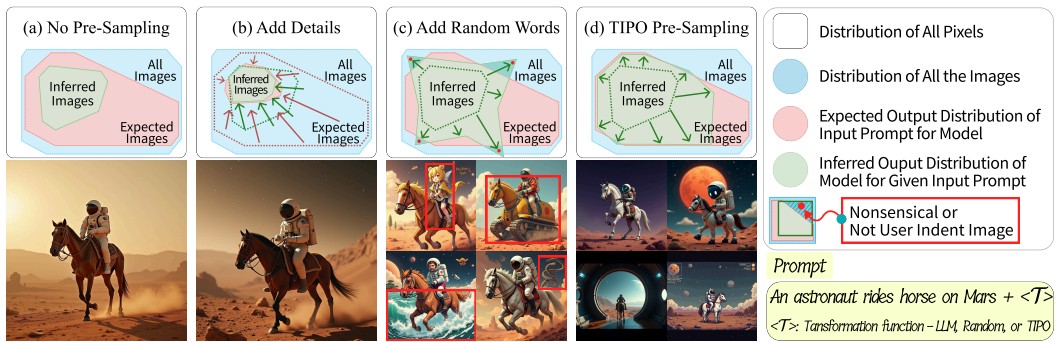

Figure 2: Illustration of various pre-sampling method for generating the T2I prompt `An astronaut rides horse on Mars` + $<\mathcal{T}>$. **(a)** yields a basic image. **(b)** enhances details of images but requires manual refinement. **(c)** adding random words may introduce irrelevant content (red boxes), exceeding the user's intent. **(d)** TIPO pre-sampling (ours) aligns outputs with expected intent, maintaining both detail and variety. $<\mathcal{T}>$ represents a transformation function for pre-sampling.

While the original image captions are fine-grained and detailed, which can yield high-quality images when all elements are used, they often result in prompts that are excessively lengthy or overloaded with information. Such prompts diverge from typical user input and pose alignment challenges. To mitigate this, we construct a simpler subset by removing some tags and sentences from the original prompt 1set, as detailed in Section 4.2.

## 4.2 FORMATTED PROMPT CONSTRUCTION

We aim to construct prompts in a unified format compatible with existing image captions. First, we incorporate the common *metadata* present in these image captions, typically represented as `<Category>: <Content>`. These metadata categories primarily include `style`, `aspect ratio`, `quality`, and `year` (e.g., `quality: masterpiece, style: Impressionist`). This structured metadata is intuitive for users to read and edit, while also providing strong guidance to downstream T2I models on the generation scope.

Next, we construct both tag-based and NL-based prompts using text sets $T$. Our design generates both simple (incomplete) and complete prompts for each image, and we train an auto-regressive language model to extend the simple prompts into complete versions. For tag-based prompts, since the tags are largely order-insensitive (i.e., the order has minimal impact on T2I outcomes), we propose a prefix-based dropout strategy. We first randomly shuffle the complete set of tags $T = \{t_1, t_2, t_3, \ldots, t_n\}$ from a given image caption. Then, we construct a simpler tag set $T_s = \{t_1, t_2, \ldots, t_m\}$ by randomly selecting $m < n$ tags. The prompts are constructed as:

$$p_s = \text{concat}(T_s), \quad p_o = \text{concat}(T)$$

Here, $p_s$ and $p_o$ denote the simple and original prompts, respectively. By ensuring that $p_s$ is always a prefix of $p_o$, the language model can readily expand simple tag-based prompts into complete versions.

For NL-based prompts, however, this strategy cannot be applied directly because the first sentence often contains crucial information (Godbole et al., 2024) and the order of sentences significantly influences the caption's semantics. Therefore, we preserve the first sentence and randomly drop some of the subsequent sentences without changing their order. Let:

$$S = [\text{sentence}_1, \text{sentence}_2, \text{sentence}_3, \ldots, \text{sentence}_n]$$

represent the ordered sequence of sentences in an image caption. We derive a simple subsequence $S_s$ by randomly selecting $m$ sentences from $S$ while ensuring that the first sentence is always included and that the original order is maintained. In other words,

$$S_s = [\text{sentence}_1, \text{sentence}_{i_2}, \ldots, \text{sentence}_{i_m}],$$

with $1 < i_2 < \ldots < i_m \leq n$ and $m < n$. The simple and complete NL-based prompts are then constructed as:

$$p_s = \text{concat}(S_s), \quad p_o = \text{concat}(S_s, S)$$

This ensures that $p_s$ remains a prefix of $p_o$. Although some sentences may be repeated in $p_o$, selecting a smaller $m$ effectively mitigates this, and it does not empirically affect the generation quality.

## 4.3 TEXT PRE-SAMPLING

We aim to reformulate user input into forms that better align with the high-quality training text distribution $p_o$ via *pre-sampling*, which stands for "text sampling before image sampling." A naïve strategy is plain text completion, where tokens are directly appended to the user input in an unstructured manner. Such completion often mirrors the low-quality phrasing of the input, deviates from the distribution of high-quality prompts, and produces inferior generations. On the other hand, a full rewrite risks deviating from the user's original intention. To balance these issues, TIPO preserves the original input and appends a structured, distribution-consistent expansion. This appended segment is typically paragraph-like or tag-like, making it both informative and easy to edit or remove (see Figure 2 for an illustration and Appendix E for concrete examples).

We propose the core technique of TIPO, a flexible pre-sampling mechanism that decomposes prompt optimization into three subtasks: enriching tag sequences, extending natural language (NL) prompts, and refining NL prompts. For example, a short NL prompt can be expanded into a detailed tag sequence (*short_to_tag*), as illustrated in Figure 3. We further distinguish between *basic tasks*, which perform a single transformation, and *composite tasks*, which chain multiple transformations within a single forward pass. The latter expose the model to more holistic training signals while reducing computational overhead. Table 1 summarizes all tasks and their input–output forms.

| Task | Description |
|---|---|
| *Basic tasks* | |
| tag_to_long | Generate a new NL prompt given tags. |
| long_to_tag | Extend a tag sequence given an NL prompt. |
| short_to_tag | Extend a tag sequence given a short/simple prompt $p_s$. |
| short_to_long | Generate a refined, detailed NL prompt given a user-provided NL prompt. |
| *Composite tasks* | |
| short_to_tag_to_long | From a short NL prompt or tags, produce a refined NL prompt. |
| short_to_long_to_tag | From a short or generated NL prompt, extend a tag sequence. |
| tag_to_short_to_long | From tags or NL prompts, generate a refined NL prompt. |

Table 1: Pre-sampling tasks in TIPO. Basic tasks focus on one-step transformations, while composite tasks combine two basic tasks within a single forward pass.

We randomly select from the aforementioned tasks during training to enhance model generalization. By extensively training on these tasks, TIPO can seamlessly adapt to various input types, flexibly refining user input whether it consists of tags, short sentences, or long sentences. Figure 3 (b) illustrates a scenario where both tag captions $T_s$ and short NL captions $S_s$ are available. In such cases, TIPO processes each input type separately to maintain clarity and coherence:

$$S_s = [\text{A young girl with long hair...}], \quad \texttt{metadata} = \emptyset$$
$$T_s = \{\text{outdoors, scenery, water, wind, landscape, \dots}\}$$

The generation proceeds sequentially as follows:

1. short_to_tag: TIPO uses $T_s$ as the primary prompt to generate a detailed tag sequence $T_d$.

2. tag_to_long: $T_d$ is incorporated into the metadata, and TIPO produces a refined short NL prompt $S_s$ based on $T_d$.

3. short_to_tag_to_long: With both $T_d$ and $S_s$ in the metadata, TIPO generates a comprehensive long NL prompt $S_d$, ensuring a more detailed output.

4. TIPO aggregates $T_d$, $S_d$, and any additional metadata to construct a context-rich prompt $p_d$.

This progressive process enables TIPO to build prompts that are both detailed and contextually aligned with the user's input.

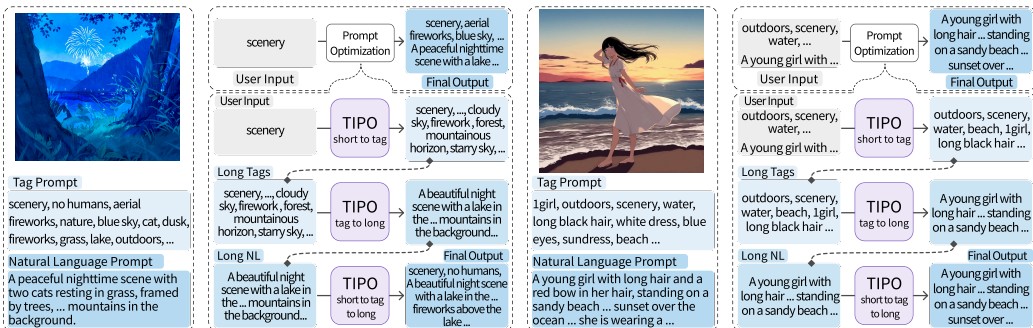

(a) Scenery tag only input           (b) Tag + short NL input

Figure 3: Example prompt optimization paths in TIPO. (a) shows generation from a single tag input, while (b) uses both tag and natural language input. These illustrate representative pipelines, not the full range of use cases. Blue shading indicates increasing prompt richness.

**Implementation Details** In implementation, TIPO adopt the LLaMA architecture (Touvron et al., 2023a;b; AI@Meta, 2024)[1], with all experiments are conducted with a 200M-parameter model [2]. Our training dataset is about 40 billion tokens curated from Danbooru2023 (nyanko202, 2023; Yeh, 2024b), GBC10M (Hsieh et al., 2024), and CoyoHD11M (CaptionEmporium, 2024).

# 5 EXPERIMENTS

## 5.1 EXPERIMENTAL SETTINGS

**Baselines and T2I Models** We compare against representative prompt-optimization baselines: GPT-4o-mini for zero-shot rewriting (OpenAI, 2024), MagicPrompt, which fine-tunes GPT-2 on community-collected prompts (daspartho, 2022), Promptist, which applies reinforcement learning for model-preferred prompt optimization (Hao et al., 2023), and Gemini-2.0-flash-image [3], which uniquely serves both as a prompt-refinement baseline and a T2I generator. For image generation backbones, our main experiments adopt *SDXL-base-1.0* (Podell et al., 2024), *Illustrious v3.5 (v-pred)*, *Kohaku-XL-Zeta*, and *Stable Diffusion 3.5 Large*(Esser et al., 2024a). To further assess generalization, we additionally evaluate on four diverse backbones with undisclosed training data: `FLUX.1-dev` (black-forest labs, 2024), `Omnigen2` (Wu et al., 2025), `Lumina-2` (Qin et al., 2025), and `HiDream-I1` (Cai et al., 2025). The details of T2I models are provided in Appendix B.

**Evaluation Metrics** We employ four latest metrics FDD (Stein et al., 2023), Aesthetic Score (discus0434, 2024), AI Corrupt Score (narugo1992, 2023), and Vendi Score (Friedman & Dieng, 2022) to measure the quality of generated images. Specifically, FDD (Fréchet DINO Distance) quantifies fidelity by comparing the distribution of DINOv2 features (Oquab et al., 2023) between reference images in the evaluation dataset and images generated from the corresponding captions, which better aligns with human perception than traditional FID (Heusel et al., 2017). Aesthetic Score is computed via Aesthetic Predictor V2.5 (discus0434, 2024), quantifying visual appeal, composition quality, and artistic merit. AI Corrupt Score detects technical flaws in generated images by identifying visual artifacts. Vendi Score quantifies image diversity by calculating the von Neumann entropy from a normalized cosine similarity matrix using DinoV2 embeddings. Notably, Vendi is defined directly on feature-space dispersion and is sensitive to non-semantic variations such as low-level noise or artifacts. As a result, it "should be used alongside a quality metric" (Friedman & Dieng, 2022) and a higher value alone does not always correspond to more meaningful semantic diversity.

---

[1]The multi-task design of TIPO is compatible with many autoregressive language models (e.g., GPT, LLaMA, Qwen, etc.). Intuitively, adopting more advanced backbones could further enhance efficiency and effectiveness, but such exploration would require extra training cost, which we defer to future work.

[2]We also trained TIPO-100M and 500M variants to analyze the impact of model scales. See Appendix C.

[3]https://developers.googleblog.com/experiment-with-gemini-20-flash-native-image-generation/

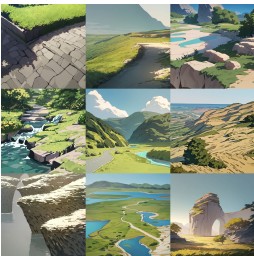 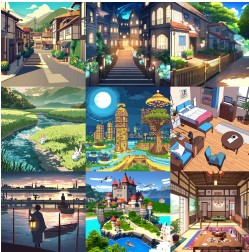 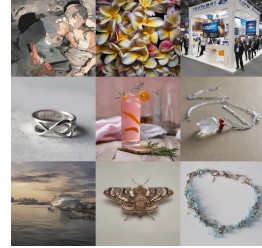 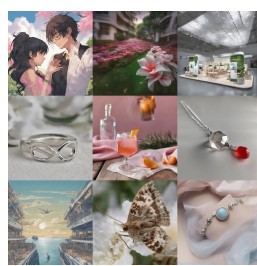

(a) Scenery Tag Only    (b) Scenery Tag + TIPO    (c) Truncated Long Prompt    (d) Truncated + TIPO

Figure 4: Generated images from 4 types of prompts: (a) simple scenery tag, (b) scenery tag enhanced by TIPO, (c) truncated (< 40 words) long prompt, (d) TIPO-enhanced truncated prompt. TIPO adds detail and maintains variety, yielding coherent images from simple prompts.

**Evaluation Protocols**    Our proposed TIPO leverages large-scale image caption datasets for training, which overlap with the training text distributions of many T2I models. Following Promptist (Hao et al., 2023), we divide our experiments into two settings: (1) **In-domain**, where the T2I model's training texts overlap with those used by TIPO, and (2) **Out-of-domain**, where no overlap exists.

## 5.2    EXPERIMENTAL RESULTS

**In-domain Tag-based Prompt Optimization**    To assess prompt optimization performance on tag-based prompts, we generate scenery images, as they contain abundant descriptive tags for objects and backgrounds. We randomly sample 32,768 tag-based prompts from Danbooru2023 (Yeh, 2024b; nyanko202, 2023), shuffle and concatenate the scenery tags into new prompts (thereby preventing data leakage of the original captions), and generate one image per prompt using `Kohaku-XL-Zeta`. This evaluation tests the *in-domain* capabilities, as Danbooru2023 is used during both TIPO and `Kohaku-XL-Zeta` training. The results in Table 2 reveal two key insights. First, MagicPrompt and Promptist, which rely on user prompts or reinforcement learning, underperform in Aesthetic and AI Corrupted Scores due to the quality or quantity limitations of their collected samples (e.g., Promptist uses 90K samples, limited by the high reinforcement learning cost). In contrast, GPT and TIPO benefit from large-scale training corpora (>30M samples), yielding higher-quality outputs. Second, TIPO achieves the best FDD by a substantial margin over GPT, which can be attributed to its superior distribution alignment with T2I models.

**In-domain NL-based Prompt Optimization**    We evaluate the prompt optimization performance on NL-based prompts by selecting 10,000 short prompts and 10,000 long prompts from CaptionEmporium (CaptionEmporium, 2024) and GBC (Hsieh et al., 2024) as test prompts. In particular, since the long prompts are much longer than typical user input, we truncate them to two sentences (< 40 words) to simulate real-world applications. We use `SDXL-1.0-base` as the T2I model, whose training text data largely overlap with TIPO. Table 2 demonstrates that TIPO achieves either the best or second-best scores in Aesthetic and AI Corrupt Score by effectively enriching the original prompt with appropriate textual elements while rarely introducing extraneous noise. While all methods compromise fidelity and diversity, as reflected in the FDD and Vendi Score, TIPO remains competitive because it maintains small semantic deviation from the original sentences via progressive refinement.

**Out-of-domain Performance**    Some recent T2I models are trained on proprietary images and captions. As a representative example, `SD-3.5-Large` is trained on private images captioned with CogView (Zheng et al., 2024), which differ markedly from the texts used to train TIPO. To evaluate model performance in this out-of-domain scenario, we generate 8,192 original tag- and NL-based prompts using the baseline GPT-4o-mini rather than relying on existing prompt datasets. We apply the remaining methods to these prompts and assess their performance.

As shown in Table 2 , SD-3.5-Large faithfully generates images that align well with GPT-produced prompts. Consequently, additional optimizations tend to reduce fidelity and introduce more artifacts. Nevertheless, GPT-generated prompts are accurate and lack diversity. TIPO optimization enriches

Table 2: Comprehensive performance comparison of TIPO against baselines across different in-domain prompt types. Metrics are marked with ↑ (higher is better) or ↓ (lower is better). For OOD tests, the 'Original' baseline and the FDD metric was not applicable. In the table, **Aesthetic** refers to Aesthetic Score, **Corrupt** to AI Corrupt Score, and **Vendi** to Vendi Score. TIPO demonstrates significant improvements, achieving the highest average rank among all baselines. Best results are in **bold** and second-best are underlined.

| Prompt Type / Task | Metric | Original | GPT | MagicPrompt | Promptist | TIPO |
|---|---|---|---|---|---|---|
| In-domain Tag-based Prompts | FDD ↓ | 0.3558 | 0.5414 | 0.3247 | 0.2350 | **0.2282** |
| | Aesthetic ↑ | 5.0569 | **6.3676** | 6.1609 | 5.9468 | 6.2571 |
| | Corrupt ↓ | 0.5743 | 0.2510 | 0.4976 | 0.4331 | **0.0805** |
| | Vendi ↑ | **16.814** | 8.663 | 11.901 | 14.327 | 13.307 |
| In-domain NL-based (Short) | FDD ↓ | **0.0957** | 0.1668 | 0.0980 | 0.1783 | 0.1168 |
| | Aesthetic ↑ | 5.8370 | **6.0589** | 5.8213 | 5.7963 | 5.8531 |
| | Corrupt ↓ | 0.2887 | 0.3015 | 0.2936 | 0.3686 | **0.2870** |
| | Vendi ↑ | **38.172** | 34.714 | 38.155 | 34.127 | 37.065 |
| In-domain NL-based (Truncated Long) | FDD ↓ | **0.0955** | 0.1683 | 0.1247 | 0.2096 | 0.1210 |
| | Aesthetic ↑ | 5.7497 | **6.0168** | 5.8191 | 5.7759 | 5.8364 |
| | Corrupt ↓ | 0.3132 | 0.3288 | 0.3259 | 0.4075 | **0.2870** |
| | Vendi ↑ | **38.253** | 34.811 | 37.841 | 33.527 | 37.090 |
| Out-of-Domain (OOD) Test | Aesthetic ↑ | N/A | **6.7125** | 6.4507 | 6.3924 | 6.0536 |
| | Corrupt ↓ | N/A | **0.0518** | 0.1423 | 0.0947 | 0.0720 |
| | Vendi ↑ | N/A | 8.9718 | 15.872 | 16.489 | **21.571** |
| Overall | Average Rank ↓ | 2.58 | 3.00 | 3.00 | 3.87 | **2.07** |

Table 3: TIPO on T2I models with undisclosed training data. Significant improvements are in **bold**.

| Model | Variant | Aesthetic ↑ | AI Corrupt ↓ | FDD ↓ | Vendi ↑ |
|---|---|---|---|---|---|
| FLUX.1-dev | Original | 5.2029 | 0.1202 | 0.1185 | 36.2597 |
| | TIPO | **5.2746** | **0.0938** | 0.1202 | 35.6489 |
| Omnigen2 | Original | 5.2629 | 0.1110 | 0.1373 | 33.7724 |
| | TIPO | 5.0661 | 0.1187 | **0.1253** | **34.2681** |
| Lumina-2 | Original | 5.2588 | 0.0981 | 0.1318 | 34.7599 |
| | TIPO | **5.4105** | **0.0916** | **0.1286** | 34.4655 |
| HiDream-I1 | Original | 5.7768 | 0.1055 | 0.1444 | 34.6318 |
| | TIPO | **5.8143** | **0.1019** | **0.1379** | 34.3544 |
| Gemini-2.0-Flash-Image | Self-refined | 5.2584 | 0.0928 | 0.8084 | 32.7422 |
| | TIPO | **5.2649** | **0.0700** | **0.7964** | 32.1136 |

the prompts with additional details that harmonize with the original themes, significantly enhancing the diversity of the generated images.

**Compatibility with other T2I Models**   In addition to the above baselines with publicly known training data, we further evaluate TIPO on recent models with undisclosed training sources: `FLUX.1-dev` (black-forest labs, 2024), `Omnigen2` (Wu et al., 2025), `Lumina-2` (Qin et al., 2025), `HiDream-I1` (Cai et al., 2025), and `Gemini-2.0-flash-image` (Google), using 1,000 prompts sampled from the COYO/GBC datasets. Notably, Gemini-2.0-flash-image can perform its own prompt optimization, making it a strong closed-source baseline. As shown in Table 3, TIPO generally improves image quality and alignment relative to the baselines. For example, `Lumina-2` and `HiDream-I1` achieve higher aesthetic scores and lower corruption rates after TIPO optimization. Despite Gemini's integrated optimization, TIPO still yields measurable improvements, highlighting the practical value of our lightweight, task-specific strategy. Overall, these results demonstrate that TIPO maintains robust compatibility with heterogeneous T2I models, even when their training data are unavailable and potentially mismatched with TIPO's optimization corpus.

**Efficiency**   A key concern is whether TIPO's iterative pre-sampling strategy introduces noticeable latency. Hence, we benchmark prompt-generation latency in Table 4, where TIPO reduces per-prompt latency with an improvement up to 29.4%. Details of training and inference are in Appendix H.4.

Table 4: Prompt generation latency and relative speedup

|  | **TIPO** | Promptist | PromptExtend | MagicPrompt |
|---|---|---|---|---|
| Avg. Time (s) | 1.03 | 1.46 | 1.38 | 1.14 |
| TIPO speedup vs. each | — | **+29.4%** | +25.6% | +9.6% |

**Human Preference Evaluation**   Quantitative metrics may not fully align with human preference. Therefore, we conducted a user study based on pairwise image comparisons between the original prompt, MagicPrompt, Promptist, and TIPO on over 1,400 images, gathering preferences from 221 volunteers. As illustrated in Figure 5a, TIPO achieved the highest overall win rate at 51.3%, significantly outperforming competitors. In out-of-domain scenarios, TIPO's win rate increased to 52.5%, demonstrating consistently strong user preference across different contexts. For further results and statistics, please refer to Appendix G.

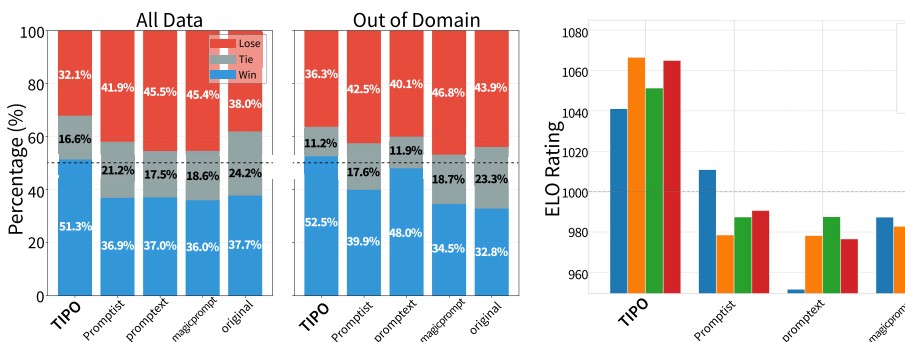

(a) Pairwise comparison results showing win-tie-lose percentages for overall user preference. Evaluations cover 'All Data' and 'Out-of-Domain' sets for TIPO and baseline methods.

(b) ELO ratings comparing TIPO and baseline methods across four criteria: Prompt Adherence, Image Quality, Aesthetic, and Overall. The dashed line indicates the initial ELO rating.

Figure 5: Human preference evaluation demonstrates that TIPO consistently achieves higher user preference compared to baseline prompt optimization methods (Promptist, prompttext, magicprompt, and original prompts). All evaluated images were generated using `SD-3.5-Medium`.

**Prompt Distribution Alignment**   While TIPO consistently achieves the best or competitive results in image quality, it remains unclear whether such gains arise from a closer distributional alignment between its optimized prompts and the T2I models' training text corpora. To verify this hypothesis, we sample from two representative T2I training sets, obtaining 1,000 natural-language captions from COYO and 1,000 tag-based captions from Danbooru2023. We then encode ground-truth captions and the outputs of all compared optimization methods using two widely adopted text encoders—T5-XXL (Raffel et al., 2020a) (used in SD3, Flux, and PixArt) and `jina-embeddings-v3` (Sturua et al., 2024) (a popular recent text-embedding model). Finally, we measure the embedding-space alignment between the ground-truth captions and optimized prompts with Fréchet Distance (FD) and Maximum Mean Discrepancy (MMD with RBF kernel). As shown in Table 5, baseline methods show varying extents of alignment across different prompt types, text encoders, and metrics, while TIPO maintains consistently better alignment under all settings. This indicates that TIPO aligns well with the T2I training-text distribution in a prompt-compatible and encoder-insensitive manner.

**Prompt–Image Alignment**   To further assess the semantic consistency between optimized prompts and their generated images, we compute CLIPScore between each prompt and its corresponding SD1.5-generated image using `openai/clip-vit-large-patch14-336`, on the same prompt sets as in the Prompt Distribution Alignment experiment. As shown in Table 6, TIPO achieves

Table 5: Embedding-space distances (FD and MMD) between optimized prompts and T2I training corpora. Lower is better. Best results are in **bold** and second-best are underlined.

| Prompt | Encoder | Metric | TIPO | Promptist | MagicPrompt | GPT-4o-mini |
|--------|---------|--------|------|-----------|-------------|-------------|
| NL-short | Jina | FD | **0.0322** | 0.1003 | 0.0385 | 0.1064 |
| | | MMD | **0.0320** | 0.1501 | 0.0624 | 0.1699 |
| | T5 | FD | **0.0704** | 0.2072 | 0.1441 | 0.1252 |
| | | MMD | **0.1438** | 0.2914 | 0.1972 | 0.2297 |
| NL-trunc | Jina | FD | **0.0309** | 0.1192 | 0.0493 | 0.0963 |
| | | MMD | **0.0359** | 0.1700 | 0.0772 | 0.1642 |
| | T5 | FD | **0.0674** | 0.2312 | 0.1884 | 0.1276 |
| | | MMD | **0.1404** | 0.3147 | 0.2270 | 0.2323 |
| Tag-based | Jina | FD | **0.1094** | 0.1891 | 0.1958 | 0.2479 |
| | | MMD | **0.1539** | 0.2473 | 0.2415 | 0.3050 |
| | T5 | FD | **0.0524** | 0.2080 | 0.2578 | 0.0728 |
| | | MMD | **0.1846** | 0.3573 | 0.3948 | 0.2194 |

Table 6: CLIPScore between optimized prompts and T2I generated images. Higher is better. Best results are in **bold** and second-best are underlined.

| Prompt Type | TIPO | MagicPrompt | GPT-4o-mini | Promptist |
|-------------|------|-------------|-------------|-----------|
| Tag-based | **0.2217** | 0.1782 | 0.1774 | 0.1642 |
| NL-short | 0.2413 | **0.2834** | 0.2378 | 0.2347 |
| NL-trunc | 0.2310 | **0.2517** | 0.2275 | 0.2063 |

the strongest alignment for tag-based prompts. While MagicPrompt shows clear advantages on NL-based prompts, this is likely due to its 1.8M+ SD1.5 community training corpus, where prompts are typically explicit and stylistically strong, effectively eliciting the CLIP encoder to produce high alignment scores. However, such stylistic bias is often less preferred by mainstream users. In contrast, TIPO attains competitive performance on NL-based prompts without such bias, as reflected by the win rate in Table 11, where MagicPrompt vs. TIPO = 11:48.

## 6 CONCLUSION

We introduced TIPO, a lightweight prompt pre-sampling framework designed for efficient real-world Text-to-Image (T2I) applications. By aligning user prompts with the intrinsic distributions of T2I training datasets, TIPO enhances semantic coherence, image fidelity, and diversity with minimal inference overhead. Experimental results show that TIPO consistently outperforms existing prompt optimization methods across multiple evaluation metrics, while extensive user studies confirm its strong alignment with human preferences. Despite these promising results, several aspects remain open for future exploration, such as generalization to out-of-distribution user input, model-specific adaptation, personalization, image-feedback-aware refinement, and large-scale model scaling. We discuss these limitations and potential extensions in Appendix J. To encourage wider adoption and facilitate reproducibility, we release our trained models and source code. We hope TIPO will inspire further advancements in efficient, scalable, and robust generative frameworks for creative systems.

## ETHICS STATEMENT

Our work democratizes prompt optimization for text-to-image models, but automatic prompt enrichment can inherit biases from training data or be misused to produce misleading or harmful content. We therefore emphasize responsible use, including bias mitigation, transparency, and appropriate user controls. No human subjects or sensitive personal data were involved, and we avoided including identifiers or metadata that could raise copyright- or attribution-related concerns.

## REPRODUCIBILITY STATEMENT

An anonymized repository with code and configuration files is provided in the supplementary materials, enabling reproduction of all reported results. The main paper and appendix reference the exact experimental settings, hyperparameters, random seeds, and evaluation scripts; data preprocessing steps and any additional resources needed to re-create the experiments are also documented there. For hardware, we report that all experiments were conducted on NVIDIA RTX 3090 and/or A6000 GPUs, with full details described in the appendix and repository.

## ACKNOWLEDGMENTS

This work was supported by Institute of Information & communications Technology Planning & Evaluation (IITP) grant funded by the Korea government (MSIT) (No. RS-2021-II211343, Artificial Intelligence Graduate School Program (Seoul National University)). This work was supported by the National Research Foundation of Korea(NRF) grant funded by the Korea government(MSIT) (No. RS-2024-00354218) This work was supported by the Technology Innovation Program(RS-2025-25456760, Development of a humanoid robot specialized in chemical processes based on AI foundation model) funded By the Ministry of Trade, Industry and Resources(MOTIR, Korea)

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

# Appendix

TABLE OF CONTENTS

## A    DATASET/RESOURCE

### A.1    DANBOORU2023

The Danbooru2023 dataset (Yeh, 2024b;a; nyanko202, 2023) is an extensive collection of images and their corresponding tags, compiled from the Danbooru image board. This dataset includes images annotated with particular and detailed tags, providing a rich resource for training both the Text-to-Image (T2I) and Large Language Models (LLMs) involved in the TIPO framework. The dataset contains data up to image ID 7,349,999, encompassing various visual content with granular annotations. These annotations allow for creating nuanced and precise prompts, ensuring that longer, more detailed prompts can indicate subsets of shorter prompts.

**Key Characteristics:**

- **Rich Annotations:** Detailed tags differentiate subtle variations, crucial for specific image generation.
- **Large Volume:** Extensive dataset size ensures diverse training examples.
- **Tag-Based Prompting:** Refined prompts from detailed tags enhance image generation accuracy.

### A.2    GBC10M

The GBC10M dataset (Hsieh et al., 2024) is a large-scale collection of 10 million images sourced from CC12M (Changpinyo et al., 2021), annotated using the Graph-Based Captioning (GBC) approach. Each image is represented by a graph where nodes correspond to object regions, compositions, and relations, and edges define their hierarchical relationships. Annotations are generated automatically through a pipeline leveraging pretrained multimodal large language models (MLLM) and object detection tools. The GBC structure enhances traditional image captions by providing detailed descriptions and structural information. Data is provided in JSON lines format, including image URLs, bounding boxes, and captions.

In TIPO, only the root node captions from GBC10M are utilized for concise yet descriptive prompts.

### A.3    COYO HD 11M

The Coyo HD 11M dataset (CaptionEmporium, 2024) consists of 11.4 million high-resolution, high-concept-density images paired with 22.8 million synthetic captions generated from the Coyo-700M dataset. Images maintain a minimum of 512 pixels on the shortest edge to ensure high visual quality. Captions, generated with the LLaVA-Next-8B model (Liu et al., 2024a) based on LLaMA 3 (AI@Meta, 2024), undergo post-processing for conciseness and clarity.

TIPO uses short and long captions, booru tags, and open image tags from this dataset.

## B    BASELINES/T2I MODELS

### B.1    PROMPT-OPTIMIZATION BASELINES

**GPT-4o-mini.**    GPT-4o-mini(OpenAI, 2024) is a multimodal model introduced by OpenAI in July 2024 as a cost-efficient variant of GPT-4o. It supports a 128k-token context window and up to 16k output tokens, trained on text and vision data. In this paper, it serves as a zero-shot rewriting baseline for prompt refinement.

**MagicPrompt.**    MagicPrompt(daspartho, 2022) fine-tunes GPT-2 models on large-scale, community-collected prompts (e.g., from Lexica.art and the Stable Diffusion Prompts dataset). It learns to generate extended prompts with richer descriptive content, originally designed to improve prompt quality for Stable Diffusion.

**Promptist.** Promptist(Hao et al., 2023) is a reinforcement learning framework for prompt optimization. Starting from a seed prompt, it generates refined prompts using supervised pre-training and reinforcement learning, guided by aesthetic and semantic alignment rewards. It consistently improves model-preferred prompt quality when paired with diffusion backbones.

**Gemini-2.0-Flash-Image.** Gemini-2.0-Flash-Image[4] is a variant of Google's Gemini 2.0 Flash family with native image generation support. It provides both prompt-refinement and text-to-image generation within a single model, supporting high-fidelity text rendering, compositional control, and iterative editing. Unlike other baselines, it functions as both optimizer and generator.

## B.2 T2I MODELS

**SDXL** Stable Diffusion XL (SDXL) (Podell et al., 2024) improves upon earlier models (Rombach et al., 2022b) with a more considerable UNet backbone and dual text encoders (CLIP ViT-L (Radford et al., 2021) and OpenCLIP ViT-bigG (Ilharco et al., 2021)), enhancing text conditioning. Supporting resolutions up to 1024×1024, SDXL accepts natural language prompts and tags, suitable for diverse image generation.

In this paper, three SDXL models are used without the refiner model:[5] SDXL-base-1.0[6], Illustrious v3.5 - vpred[7], and Kohaku-Zeta[8].

**Illustrious** Illustrious is a series of fine-tuned Stable Diffusion XL models primarily trained on the Danbooru2023 dataset. In this study, we specifically employ the v3.5 version variant with v-parameterization (Salimans & Ho, 2022), which is notable for its extensive incorporation of natural language prompts. The inclusion of both tag-based and natural language formats allows Illustrious to leverage a broad range of semantic knowledge for image generation.

Within TIPO, we perform an ablation study to analyze the effectiveness of different prompting strategies—namely extended tags versus natural language prompts—to identify which approach contributes most significantly to enhanced image generation performance.

**Stable Diffusion 3.5** Stable Diffusion 3.5 (SD-3.5) incorporates the MMDiT architecture (Esser et al., 2024a) and the Rectified Flow formulation (Liu et al., 2022; Albergo & Vanden-Eijnden, 2022; Lipman et al., 2022) for improved text-to-image generation. Utilizing triple text encoders (CLIP/ViT-L, OpenCLIP/ViT-G, T5-XXL (Raffel et al., 2020b)), SD-3.5 supports resolutions up to 1024×1024 and uses a 50/50 mix of original and CogVLM-generated captions. Figures confirm the capability to process both natural language prompts and tags.

This study employs SD-3.5-Large[9] (8B parameters) with FP8 inference on RTX 3090 or RTX 4090 GPUs.

**FLUX.1-dev** FLUX.1-dev is an open-weights text-to-image model released by Black Forest Labs as a guidance-distilled variant of their FLUX.1 family, targeting research and non-commercial use (black-forest labs, 2024). Architecturally, FLUX adopts a transformer-based rectified-flow formulation (Lipman et al., 2022) with a hybrid stack of multimodal/parallel diffusion transformer blocks and rotary position encodings, scaled to ~12B parameters, emphasizing prompt adherence, typography, and aspect-ratio flexibility (black-forest labs, 2024). We use the `FLUX.1-dev` checkpoint for T2I without additional refiners; it accepts both natural-language prompts and tag-like inputs and supports resolutions in the 0.1–2.0 MP range.[10]

**OmniGen2** OmniGen2 is a unified multimodal generator that decouples autoregressive text modeling from diffusion-based image generation via two distinct pathways with unshared parameters

---

[4]`https://developers.googleblog.com/experiment-with-gemini-20-flash-native-image-generation/`
[5]`https://huggingface.co/stabilityai/stable-diffusion-xl-refiner-1.0`
[6]`https://huggingface.co/stabilityai/stable-diffusion-xl-base-1.0`
[7]`OnomaAIResearch/Illustrious-xl-early-release-v0`
[8]`https://huggingface.co/KBlueLeaf/Kohaku-XL-Zeta`
[9]`https://huggingface.co/stabilityai/stable-diffusion-3.5-large`
[10]`https://huggingface.co/black-forest-labs/FLUX.1-dev`

(Wu et al., 2025). The diffusion side conditions on hidden states from the MLLM while *exclusively* feeding VAE features into the diffusion decoder to preserve low-level fidelity; a 3D rotary scheme (Omni-RoPE) disentangles sequence ID and 2D spatial coordinates to stabilize editing and in-context generation (Wu et al., 2025). Beyond standard T2I, OmniGen2 natively supports image editing and subject-driven in-context generation, and introduces a reflection mechanism/dataset to iteratively refine outputs.

**Lumina-Image 2.0**   Lumina-Image 2.0 proposes a unified and efficient T2I framework built on *Unified Next-DiT*—a single-stream DiT that performs joint self-attention over text and image tokens—paired with a task-tailored *Unified Captioner* (UniCap) that produces multi-granularity, multi-lingual captions for training (Qin et al., 2025). The model employs Multimodal RoPE, progressive resolution training (256→1024), and efficient inference (CFG-Renorm/Trunc (Lin et al., 2024; Yi et al., 2024), Flow-DPM-Solver (Xie et al., 2024), TeaCache (Liu et al., 2025)) to improve prompt-following and speed at only ∼2.6B parameters (Qin et al., 2025).

**HiDream-I1**   HiDream-I1 is a 17B-parameter image foundation model based on a *sparse* Diffusion Transformer with dynamic Mixture-of-Experts (MoE) (Cai et al., 2025). It employs a dual-stream (text/image) sparse DiT for separate encoding followed by a single-stream sparse DiT to fuse modalities efficiently; hybrid text encoders (e.g., CLIP-L/14, CLIP-G/14, T5-XXL) and an LLM aggregator provide robust conditioning (Cai et al., 2025). The suite includes `I1-Full` (50+ steps), `I1-Dev` (guidance-distilled, 28 steps), and `I1-Fast` (14 steps), with a GAN-powered diffusion distillation to retain sharpness at low step counts.

## C   TIPO IMPLEMENTATION DETAILS

In this appendix, we provide all the necessary details including our dataset construction process, model configurations, inference pipeline, and the model's properties not mentioned in Section 4.2 and 4.3.

### C.1   TIPO TRAINING DATA CONSTRUCTION

This section details our methodology for constructing and preprocessing training data to ensure robust model performance across various input scenarios.

**Length Control**   To systematically control output prompt length, we implement a structured length categorization system using unique length tags. These tags enforce specific constraints on tag counts and natural language sentence lengths. For instance, the `<long>` tag specifies that the corresponding prompt must contain between 36 and 52 tags (inclusive), accompanied by 4 to 8 sentences of natural language description. We define four distinct length categories, each with strict bounds for tag count and sentence length.

| Type | Very Short | Short | Long | Very Long |
|------|-----------|-------|------|-----------|
| Tags (count) | 18 | 36 | 48 | 72 |
| NL (sentences) | 2 | 4 | 8 | 18 |

Table 7: Maximum length specifications for each category and caption type. For each category, the actual count/length must not exceed these values.

**Random Augmentation**   To enhance input diversity and better simulate real-world usage patterns, we implement several data augmentation strategies:

- **Metadata Tags:** For tags representing image metadata (e.g., artist, character, aspect ratio), we employ two randomization techniques:
  - Random removal of metadata tags
  - Random repositioning of metadata tags to the end of the prompt, after all content-related descriptions

| | TIPO-100M | TIPO-200M stage1 | TIPO-200M stage2 | TIPO-500M |
|---|---|---|---|---|
| Architecture | | LLaMA | | |
| Type | Pretrain | Pretrain | Finetune | Pretrain |
| Vocab Size | | 32013 | | |
| Hidden Dim | 640 | 768 | - | 1280 |
| Attention Heads | 10 | 12 | - | 20 |
| MLP Dim | 2240 | 2304 | - | 3840 |
| Hidden Layers | 10 | 20 | - | 20 |
| Model Parameters | 100M | 203M | - | 508M |
| Max Learning Rate | 5e-4 | 2e-4 | 5e-5 | 2e-4 |
| Optimizer | | AdamW | | |
| LR scheduler | | Cosine Annealing LR | | |
| betas | | 0.9, 0.98 | | |
| weight decay | | 0.01 | | |
| Dataset | Coyo, GBC, Dan | GBC, Dan | Coyo, GBC, Dan | Coyo, GBC, Dan |
| Epoch | 1 | 5 | 3 | 5 |
| max context length | 512 | 512 | 1024 | 1024 |
| global batch size | 1024 | 2048 | 2048 | 3584 |
| Token Seen | 6.0240B | 22.625B | 18.339B | 31.274B |
| Hardware | 4 × RTX3090 | 4 × RTX3090 | 4 × RTX3090 | 8 × H100 |
| Training Time (wall) | 22.5 hour | 150 hour | 270 hour | 100 hour |

Table 8: Training settings for TIPO models. The datasets include CoyoHD11M (Coyo), GBC10M (GBC), and Danbooru2023 (Dan). Stage 2 additionally incorporates Pixtral (Agrawal et al., 2024) to generate NL captions from Danbooru2023 dataset.

This approach encourages the model to handle varying metadata positions and availability, while maintaining the ability to infer metadata relationships from content descriptions.

- **Content Tags:** For tags describing image content (e.g., objects, actions, attributes), we implement:
  - Random shuffling of tag order within the content section
  - Length-based truncation to meet target length constraints while preserving key content information
- **Natural Language:** For natural language descriptions exceeding length limitations, we employ selective sentence removal, targeting middle sentences to preserve context-setting opening sentences and concluding details. This maintains coherent narrative flow while meeting target length requirements.

These augmentation strategies create a more diverse training dataset that better reflects real-world prompt variations, improving the model's robustness and adaptability to different input styles and formats.

## C.2 TIPO TRAINING SETTINGS AND MODEL CONFIGURATIONS

**Tokenizer and Task Tokens** TIPO employs a vocabulary derived from LLaMA2 (Touvron et al., 2023b) consisting of 32,000 tokens, with additional tokens (13 tokens) specifically designated for task and length control or placeholders. This extended vocabulary includes task identifiers and length modifiers to ensure flexibility across different prompt types:

- **Placeholder Token (1 token):**

    ```
    <|empty|>
    ```

- **Task Tokens (8 tokens):**

    ```
    <|gen_meta|>, <|tag_to_long|>, <|short_to_tag|>,
    <|long_to_tag|>, <|short_to_long|>, <|short_to_tag_to_long|>,
    <|short_to_long_to_tag|>, <|tag_to_short_to_long|>
    ```

- **Length Tokens (4 tokens):**

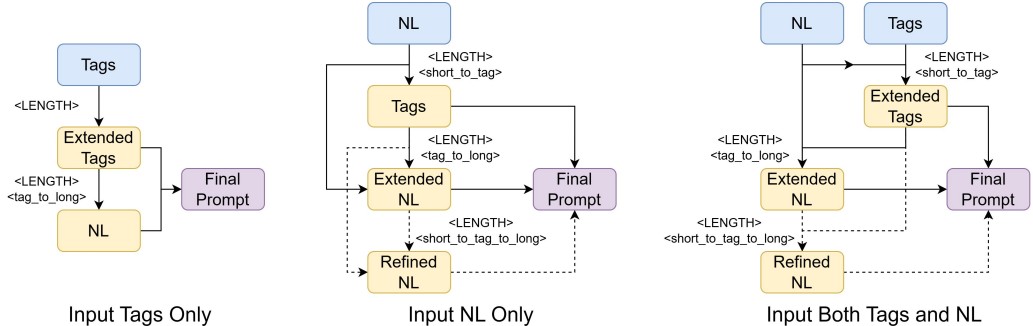

Figure 6: TIPO inference workflow, with solid arrows denoting the primary generation steps and dashed arrows indicating alternative generation paths within the same cycle. `<TOKEN>` represents special tokens, with all tokens detailed in Section C.2.

<|very_short|>, <|short|>, <|long|>, <|very_long|>

**Optimizer and Learning Schedule** Training is performed using the AdamW optimizer (Loshchilov, 2017), with a cosine annealing learning rate scheduler (Loshchilov & Hutter, 2017). The optimizer parameters include $\beta_1 = 0.9$, $\beta_2 = 0.98$, and a weight decay of 0.01. Maximum learning rates are adjusted per model size, as outlined in Table 8.

**Training Configurations** TIPO models are trained in multiple stages. Table 8 summarizes the configurations for pretraining and fine-tuning TIPO-100M, TIPO-200M, and TIPO-500M. Both pretraining and fine-tuning was conducted on datasets like Danbooru2023 (nyanko202, 2023), GBC10M (Hsieh et al., 2024), and CoyoHD11M (CaptionEmporium, 2024).

**Augmented Task Representation** Each dataset entry undergoes random task assignment and splitting to simulate a wide range of input-output mappings, effectively increasing the dataset size. For example, a single entry may contribute to tasks like `short_to_tag` or `tag_to_long`, with length modifiers dynamically controlling the output verbosity. This approach ensures the model can handle diverse tasks while maintaining robust generalization.

**Hardware and Time Requirements** Training was conducted on NVIDIA RTX3090 GPUs for smaller models and H100 GPUs for TIPO-500M. Total wall-clock training times ranged from 22.5 hours for TIPO-100M to 270 hours for fine-tuning TIPO-200M.

**Token Seen and Effective Training** Non-padding tokens are used to measure the effective token count during training, ensuring efficiency given the short and variable data lengths. Table 8 details the total tokens seen per model and training stage, illustrating the comprehensive exposure to diverse data entries.

## C.3 TIPO INFERENCE SETTINGS

**Sampling Strategy** We employ a hybrid stochastic decoding strategy combining nucleus sampling (top-$p = 0.95$) and top-$k = 60$ filtering. This follows standard practice in open-ended text generation, as adopted in the official Hugging Face generation examples [11]. This hybrid approach maintains diversity while preserving coherence, preventing both overly deterministic and excessively noisy generations.

**Inference Pipeline** The TIPO inference pipeline is designed to handle various input types and scenarios, combining different tasks to refine or expand both tag-based and natural language prompts. Figure 6 illustrates this comprehensive workflow. Our framework processes tags and natural language

---

[11]Hugging Face. "Usage — transformers 2.11.0 documentation." Example of text generation with XLNet uses top-$p = 0.95$ and top-$k = 60$. https://huggingface.co/transformers/v2.11.0/usage.html

inputs separately, allowing for specialized handling of each input type. This flexible pipeline allows TIPO to adapt to various input scenarios, whether the user provides tags, natural language descriptions, or both. By leveraging different task combinations, TIPO ensures that tag-based and natural language prompts are optimized, resulting in more detailed and effective input for text-to-image models.

## C.4 Impact of Model Size on Performance

To analyze the impact of model scales on prompt-optimization performance, we compare TIPO-200M and TIPO-500M using a 1,000-image subsample from the COYO and GBC datasets. Results are shown in Table 9.

Table 9: Prompt optimization performance of TIPO-200M and TIPO-500M on a 1k subsample.

| Metric | Task | TIPO-200M | TIPO-500M |
|---|---|---|---|
| FDD ($\downarrow$) | NL-short | 0.1529 | **0.1356** |
| | NL-long | 0.1650 | **0.1398** |
| Aesthetic ($\uparrow$) | NL-short | $5.8531 \pm 0.7501$ | $\mathbf{5.8943 \pm 0.7064}$ |
| | NL-long | $5.8364 \pm 0.7501$ | $\mathbf{5.9030 \pm 0.7015}$ |
| AI Corrupt ($\downarrow$) | NL-short | $0.2870 \pm 0.4167$ | $0.2891 \pm 0.4189$ |
| | NL-long | $0.2870 \pm 0.4150$ | $0.2862 \pm 0.4151$ |

Overall, TIPO-500M shows consistent gains in FDD and Aesthetic scores, while performance on AI Corrupt remains comparable. However, the larger 500M variant entails substantially higher computational cost without delivering proportionally greater benefits, which limits its practicality for community use; hence, all main experiments are conducted with TIPO-200M.

## C.5 Impact of Model Size on Inference Speed

We conducted comprehensive speed tests of our 100M, 200M and 500M parameter models using the inference pipeline described in Section 5.2. Each prompt requires two sequential generation steps. Our primary metric is average tokens generated per second, which reflects real-world task performance rather than theoretical maximum throughput.

The evaluation was performed using llama.cpp (Gerganov, 2023), an efficient C++ implementation that provides optimized support for various hardware accelerators, including CUDA, HIP, and Apple Metal.

| Hardware Platform | TIPO-100M | | TIPO-200M | | TIPO-500M | |
|---|---|---|---|---|---|---|
| | tok/sec | gen time | tok/sec | gen time | tok/sec | gen time |
| M1 Max (32 GPU cores) | 339.4 | 0.66 | 190.0 | 1.23 | 119.4 | 2.02 |
| RTX 3090 | 558.5 | 0.42 | 341.4 | 0.69 | 289.8 | 0.81 |
| RTX 4090 | 742.9 | 0.29 | 454.5 | 0.51 | 359.7 | 0.63 |

Table 10: Model performance comparison across different hardware platforms. Tokens per second (tok/sec) represents the average generation speed, while generation time (gen time) shows the average time in seconds required for a complete two-step prompt optimization process.

## D    EVALUATION STATISTICS

In this appendix, we provide more statistics for the result obtained in Section 5.

### D.1    IN-DOMAIN TEST REGARDING SCENERY TAG

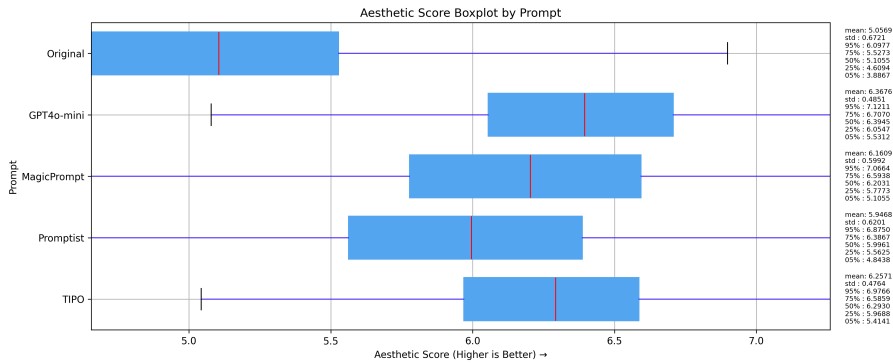

(a) The box plot for the Aesthetic Score result of scenery tag test.

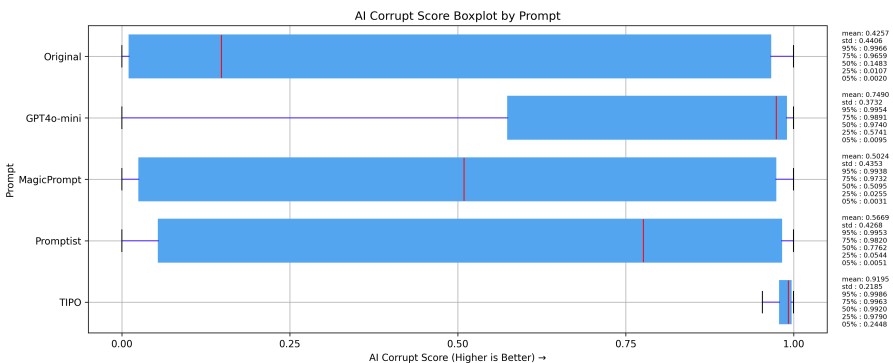

(b) The box plot for the AI Corrupt Score result of scenery tag test.

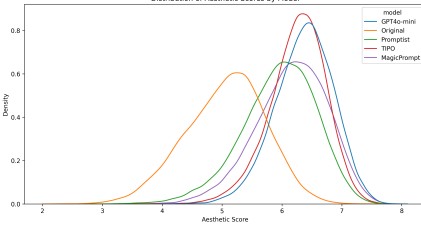

(c) The KDE plot for the Aesthetic Score result of scenery tag test.

(d) The KDE plot for the AI Corrupt Score result of scenery tag test.

Figure 7: The distribution of aesthetic and AI corrupt score for scenery tag test.

The box plot and Kernel Density Estimation (KDE) plot displayed in Figure 7 illustrate the aesthetic scores and AI corruption scores from the scenery tag test described in Section 5.2. The analysis shows that TIPO significantly outperforms all other methods, demonstrating a considerable margin of improvement.

## D.2 IN-DOMAIN PROMPT GENERATION TEST

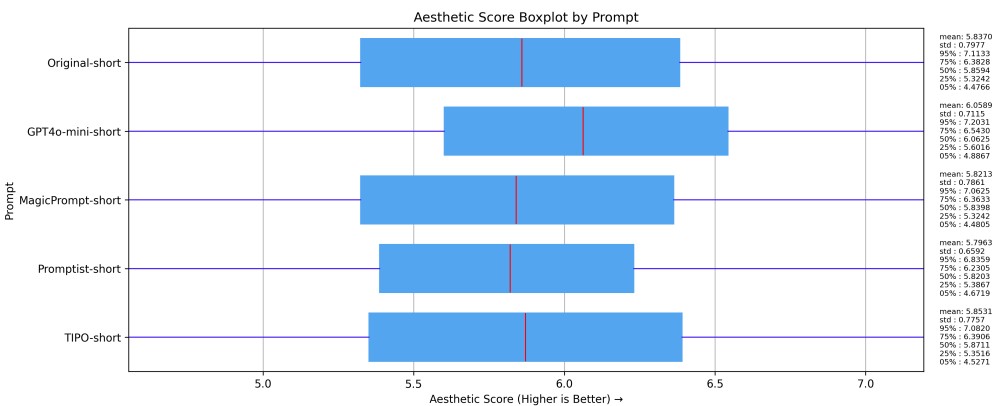

(a) The box plot for the Aesthetic Score result of short prompt input.

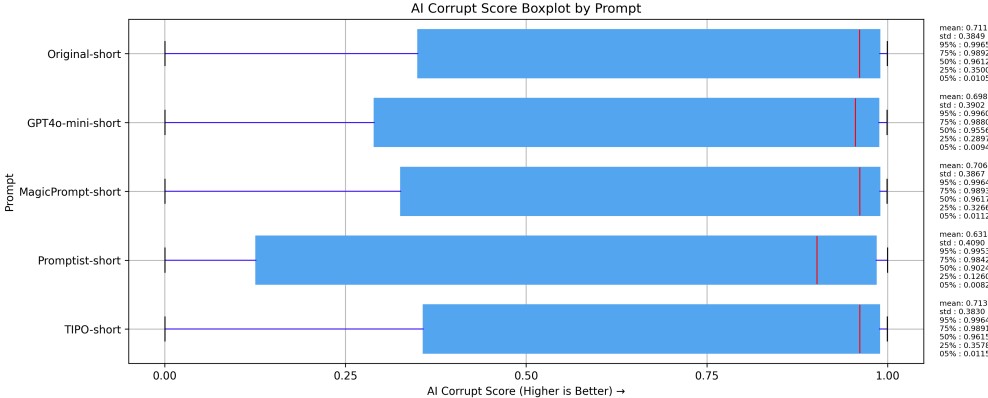

(b) The box plot for the AI Corrupt score result of short prompt input.

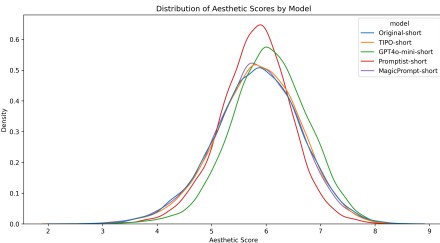

(c) The KDE plot for the Aesthetic Score result of short prompt input.

(d) The KDE plot for the AI Corrupt Score result of short prompt input.

Figure 8: The distribution of aesthetic and AI corrupt score for short prompt input.

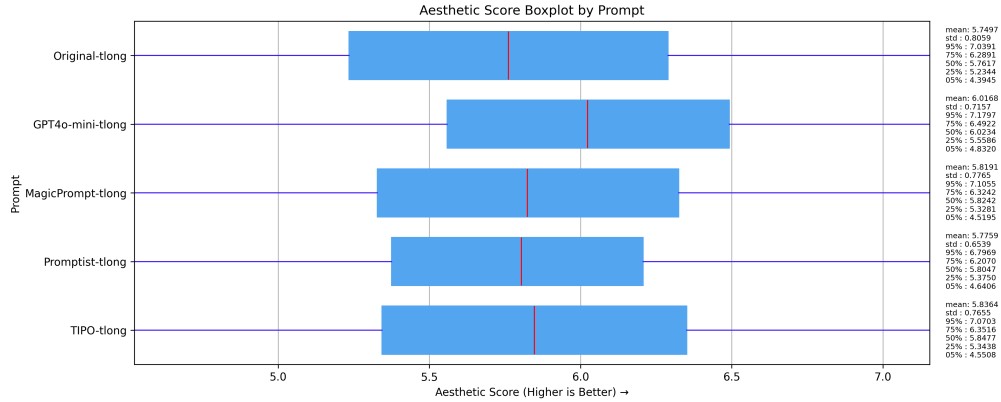

(a) The box plot for the Aesthetic Score result of truncated long prompt input.

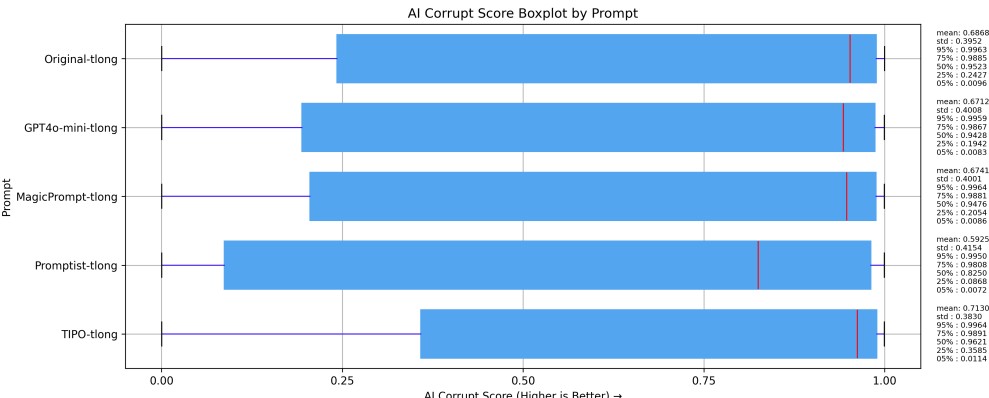

(b) The box plot for the AI Corrupt score result of truncated long prompt input.

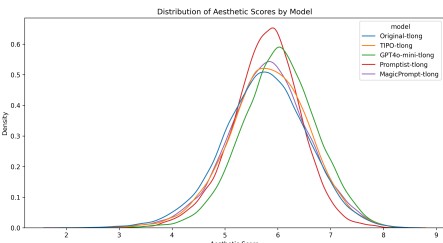

(c) The KDE plot for the Aesthetic Score result of truncated long prompt input.

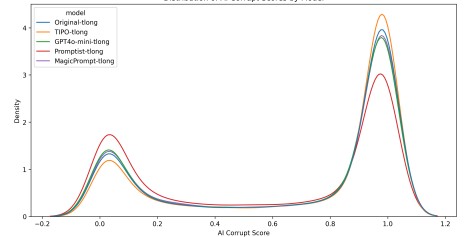

(d) The KDE plot for the AI Corrupt Score result of truncated long prompt input.

Figure 9: The distribution of aesthetic and AI corrupt score for truncated long prompt input.

Figures 8 and 9 display the box plots and KDE plots of aesthetic scores and AI corruption scores obtained from the In-domain prompt generation test detailed in Section F.2. While the box plots reveal subtle differences in performance between various methods, the AI corruption scores provide valuable insights. Specifically, these scores indicate that implementations supported by TIPO produce more stable output images than other methods.

## D.3 OUT-OF-DOMAIN EVALUATION

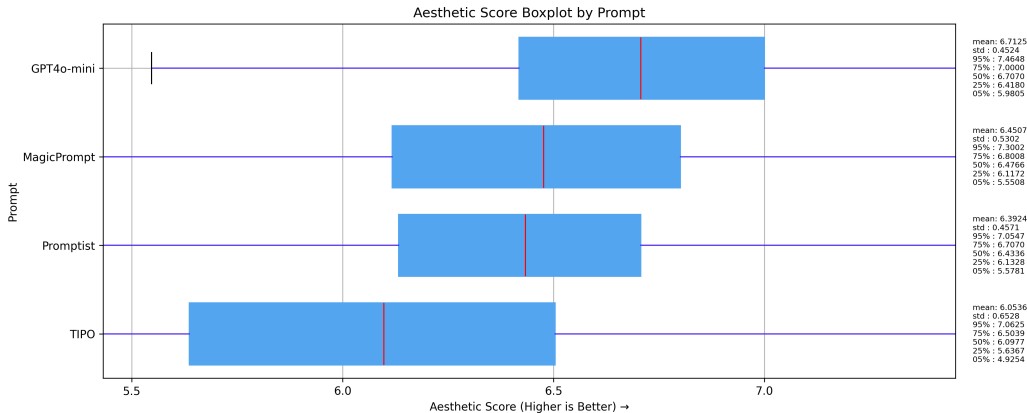

(a) The box plot for the Aesthetic Score result of out-of-focus test.

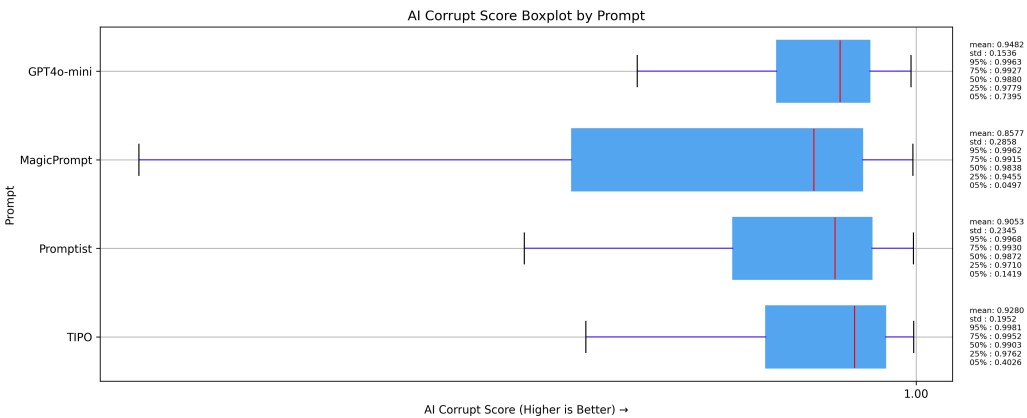

(b) The box plot for the AI Corrupt Score result of out-of-focus test.

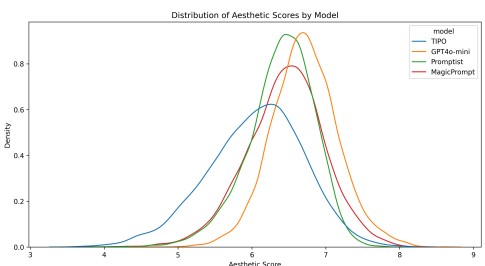

(c) The KDE plot for the Aesthetic Score result of out-of-focus test.

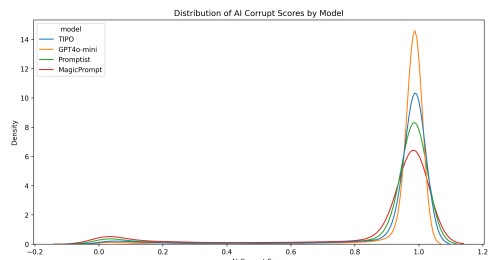

(d) The KDE plot for the AI Corrupt Score result of out-of-focus test.

Figure 10: The distribution of aesthetic and AI corrupt score for out-of-focus test.

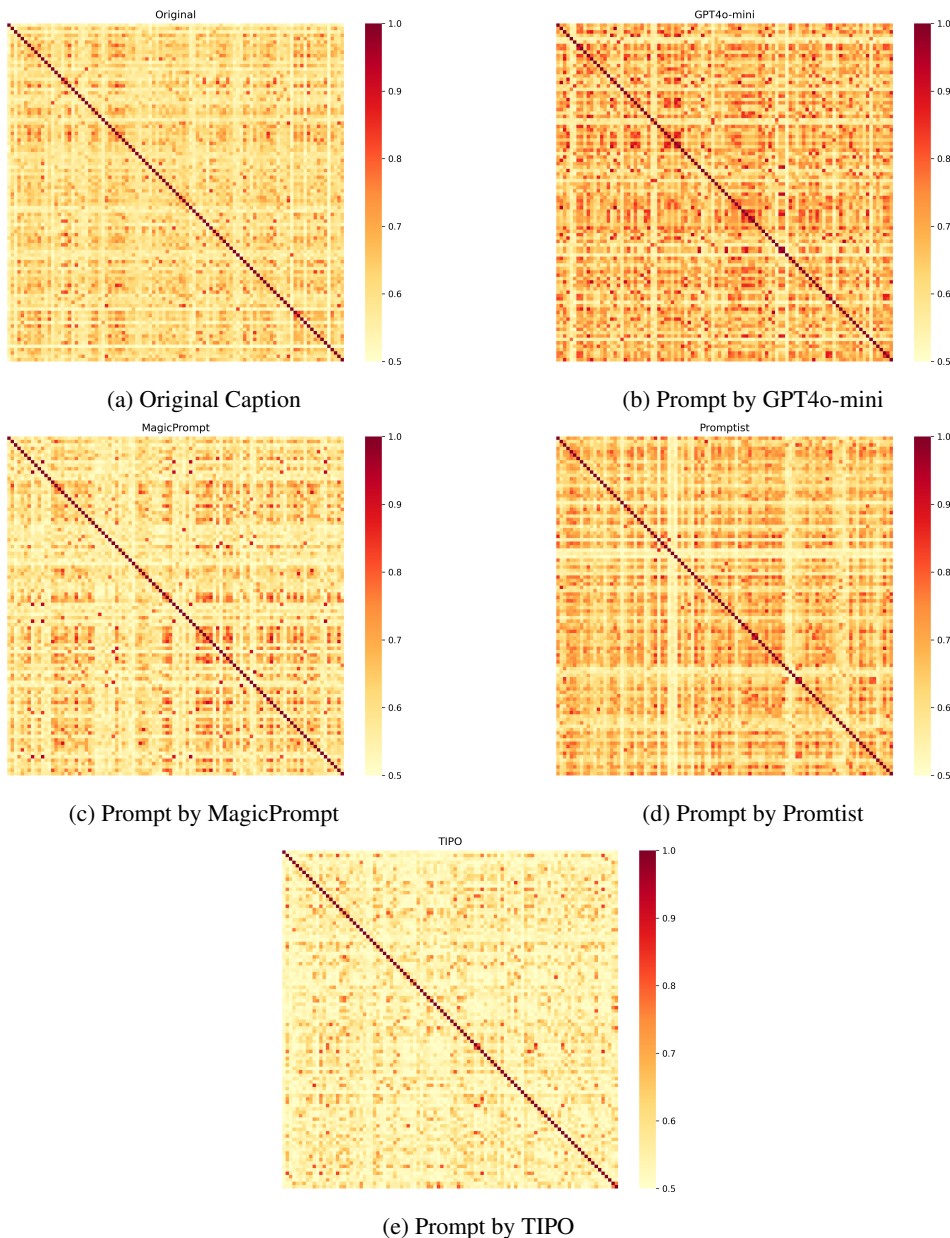

(a) Original Caption

(b) Prompt by GPT4o-mini

(c) Prompt by MagicPrompt

(d) Prompt by Promtist

(e) Prompt by TIPO

Figure 11: The similarity matrix for the 100 best aesthetic results generated in the SD3.5-Large experiments. Off-diagonal elements of the matrix indicate the similarity between different images. A lower value for an off-diagonal element indicates greater diversity among the generated images.

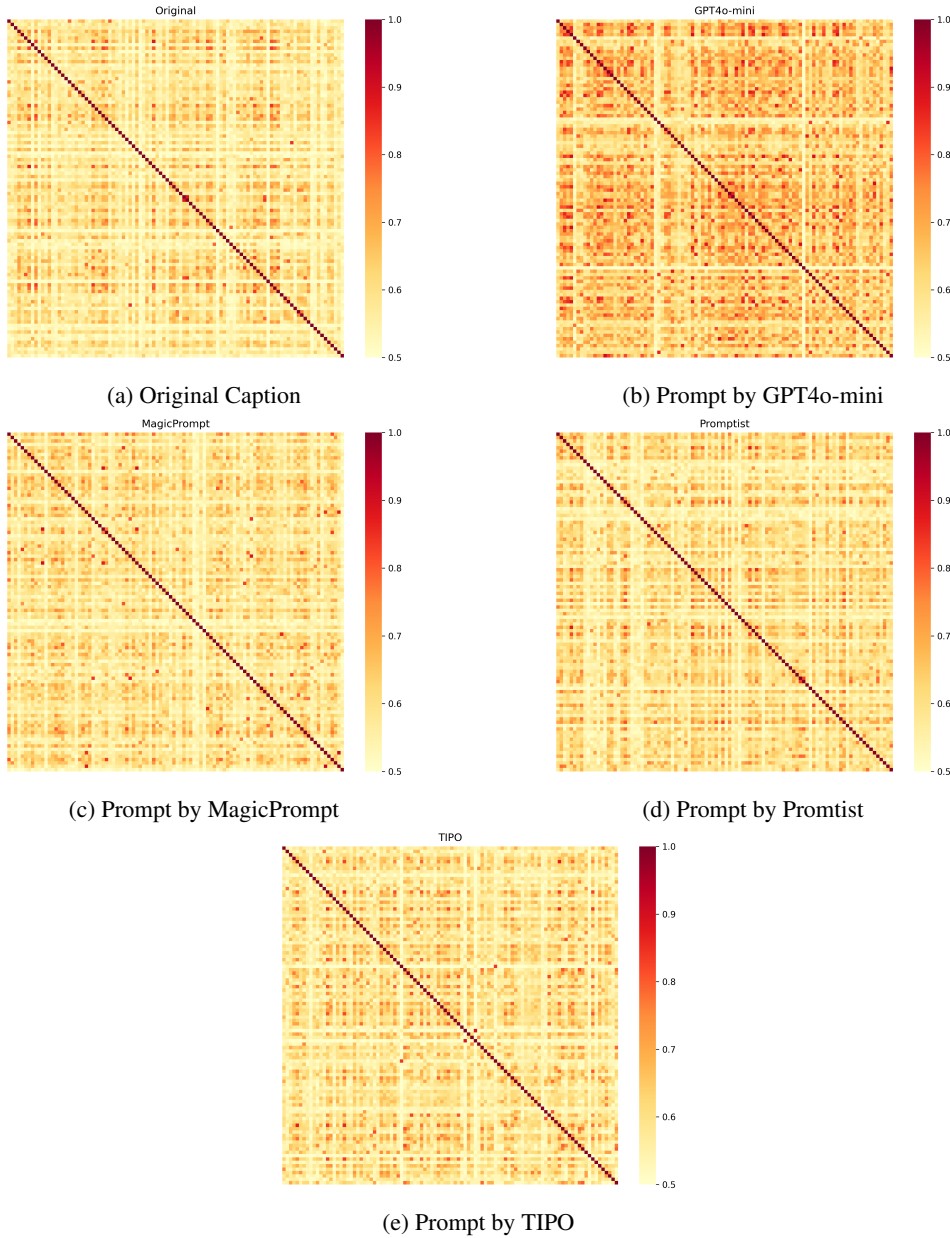

Figure 12: The similarity matrix between 100 images of worst aesthetic generated results of SD3.5-Large experiments.

Figures 11 and 12 present similarity matrices for different prompt generation methods and their corresponding aesthetic outputs on SD3.5-Large (Esser et al., 2024b). A matrix with predominantly lower similarity values (brighter appearance) indicates high diversity among generated images, while higher values (darker appearance) suggest consistent but less diverse outputs. Please refer to Table 2 in Section 5.

## D.4    ABLATION TEST

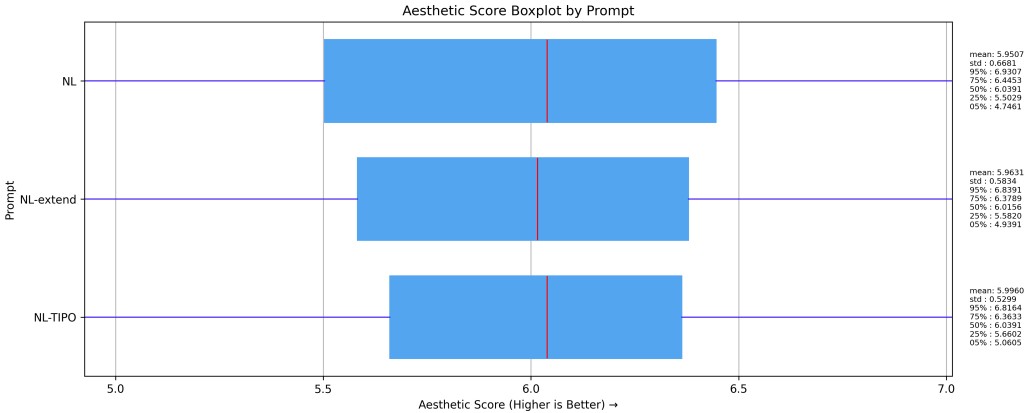

(a) The box plot for the Aesthetic Score result of NL ablation test.

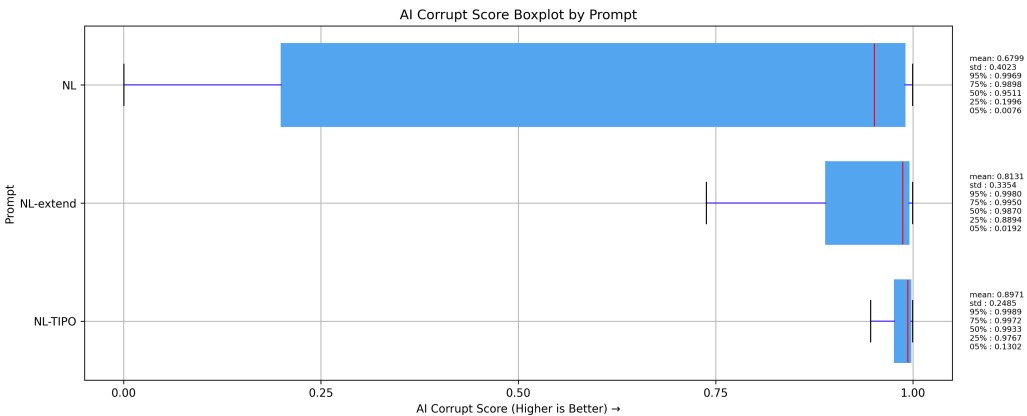

(b) The box plot for the AI Corrupt Score result of NL ablation test.

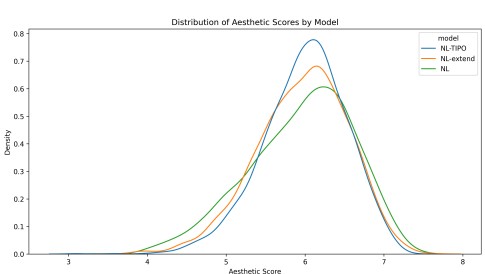

(c) The KDE plot for the Aesthetic Score result of NL ablation test.

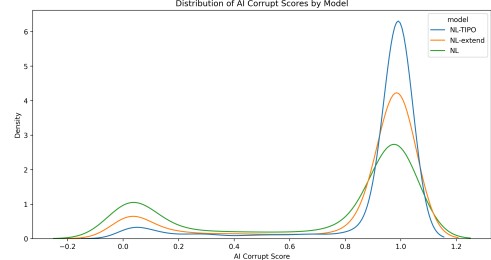

(d) The KDE plot for the AI Corrupt Score result of NL ablation test.

Figure 13: The distribution of aesthetic and AI corrupt score for NL ablation test.

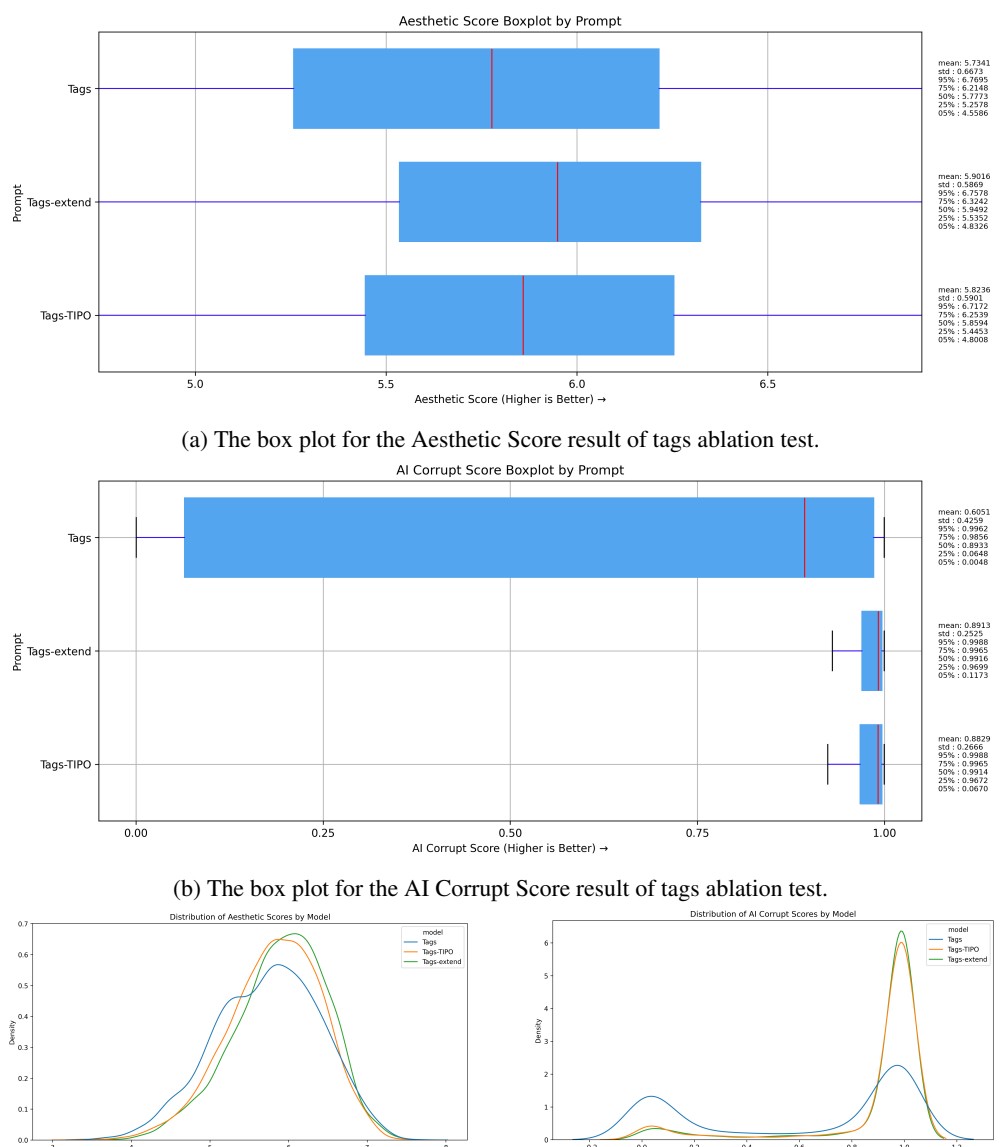

(a) The box plot for the Aesthetic Score result of tags ablation test.

(b) The box plot for the AI Corrupt Score result of tags ablation test.

(c) The KDE plot for the Aesthetic Score result of tags ablation test.

(d) The KDE plot for the AI Corrupt Score result of tags ablation test.

Figure 14: The distribution of aesthetic and AI corrupt score for tags ablation test.

Figures 14 present the tag ablation test in the TIPO effect on the aesthetic score and AI Corrupt Score among the original tag, tag-extend and the tags TIPO. The box plot reveals that the tag TIPO is better than the original tag and the tag extend is the best. In detail, KDE plot reveals that the tag TIPO has a similar performance compared with the tag extend. Both of them are better than the original tag, which indicates that the tag TIPO aspect helps control corruption and promotes the aesthetic score.

# E    TIPO EXAMPLE

In this section, we provide some text example of TIPO's input and output.

---

**TIPO Format Template**

**User Input:**
```
1girl, ciloranko, maccha (mochancc), ningen mame, ask (askzy), solo, masterpiece,
    absurdres, newest, safe

A girl sits in a cozy cafe, cradling a cup of coffee in her hand
```

- - - - - - - - - - - - - - - - - - - - - - - - - - - - - - - - - - - - - - - - - - - -

**Formatted TIPO Input for Expand Tags:**
```
meta: absurdres
rating: safe
style: anime-style illustration, digital art, character design, fantasy concept art
quality: masterpiece, newest
aspect_ratio: 1.0
target: <|short|> <|short_to_tag|>
short: A girl sits in a cozy cafe, cradling a cup of coffee in her hand
tag: 1girl, solo
```

- - - - - - - - - - - - - - - - - - - - - - - - - - - - - - - - - - - - - - - - - - - -

**Formatted TIPO Output after Expand Tags and Expand Natural Prompt:**
```
meta: absurdres
rating: safe
style: anime-style illustration, digital art, character design, fantasy concept art
quality: masterpiece, newest
aspect_ratio: 1.0
target: <|short|> <|tag_to_long|>
tag: 1girl, solo, sitting, closed mouth, jewelry, long hair, looking at viewer, crossed
    legs, plant, table, couch, bracelet, cup, smile, teacup, indoors, blue eyes,
    blonde hair, holding
long: A girl sits in a cozy cafe, cradling a cup of coffee in her hand. The cafe has
    large windows with green plants on the walls and a wooden table in front of her.
    Behind her is a staircase leading to another room. The overall atmosphere of the
    image is serene and inviting.
```

- - - - - - - - - - - - - - - - - - - - - - - - - - - - - - - - - - - - - - - - - - - -

**Formatted Output for Text-to-Image:**
```
1girl, ciloranko, maccha (mochancc),
    ningen mame, ask (askzy),

solo, sitting, closed mouth, jewelry,
    long hair, looking at viewer, crossed
     legs, plant, table, couch, bracelet,
     cup, smile, teacup, indoors, blue
     eyes, blonde hair, holding,

A girl sits in a cozy cafe, cradling a
     cup of coffee in her hand. Behind her
     is a staircase leading to another
     room. The cafe has large windows with
     green plants on the walls and a
     wooden table in front of her. The
     overall atmosphere of the image is
     serene and inviting.

masterpiece, newest, absurdres, safe
```

User Input        TIPO Output

Figure 15: An example of formatted content used for training and inference in TIPO.

---

**TIPO Format template**

**User Input:**

scenery, no humans, masterpiece, absurdres, newest, safe

- - - - - - - - - - - - - - - - - - - - - - - - - - - - - - - - - - - - - - - - - -

**Formatted TIPO Input For Expand Tags:**

```
meta: absurdres
rating: safe
quality: masterpiece, newest
aspect_ratio: 1.0
target: <|long|>
tag: scenery, no humans
```

- - - - - - - - - - - - - - - - - - - - - - - - - - - - - - - - - - - - - - - - - -

**Formatted TIPO Output after Expand Tags and tag_to_long task:**

```
meta: absurdres
rating: safe
quality: masterpiece, newest
aspect_ratio: 1.0
target: <|long|> <|tag_to_long|>
tag: scenery, no humans, storefront, motor vehicle, road sign, power lines, plant,
    railing, flower pot, vanishing point, outdoors, sign, potted plant, sidewalk,
    awning, tree, bicycle, window, railroad crossing, bush, building, utility pole,
    lamppost, shop, truck, traffic light, fence, chinese text, stairs, door, bicycle
    basket, town, day, streetcar (cafe), lamp, road
long: A small town with a variety of buildings and houses. the sky is blue and there
    are trees in the background. on the left side of the image, there is an orange
    building with a sign that reads "chinese restaurant". on the right side, there are
     several other buildings with different types of shops and restaurants. in front
    of the buildings, there appears to be a street with cars parked along the road.

in the center of the illustration, we can see a train crossing signal with two red
    lights and a blue sky above it. there is also a yellow building with white walls
    and a green roof. on top of the traffic light pole, there seems to be an air
    conditioning unit. the street is lined with trees and bushes, and there is
    graffiti on the ground.
```

- - - - - - - - - - - - - - - - - - - - - - - - - - - - - - - - - - - - - - - - - -

**Formatted Output For Text-to-Image**

```
scenery, no humans, storefront, motor vehicle, road sign, power
    lines, plant, railing, flower pot, vanishing point, outdoors
    , sign, potted plant, sidewalk, awning, tree, bicycle,
    window, railroad crossing, bush, building, utility pole,
    lamppost, shop, truck, traffic light, fence, chinese text,
    stairs, door, bicycle basket, town, day, streetcar \(cafe\),
     lamp, road,
```

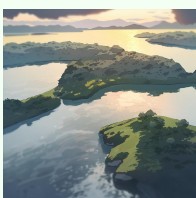

User Input

```
A small town with a variety of buildings and houses. the sky is
    blue and there are trees in the background. on the left side
     of the image, there is an orange building with a sign that
    reads "chinese restaurant". on the right side, there are
    several other buildings with different types of shops and
    restaurants. in front of the buildings, there appears to be
    a street with cars parked along the road. in the center of
    the illustration, we can see a train crossing signal with
    two red lights and a blue sky above it. there is also a
    yellow building with white walls and a green roof.

masterpiece, newest, absurdres, safe
```

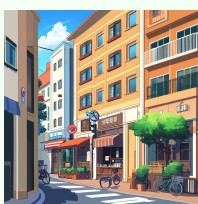

TIPO Output

Figure 16: An example formatted content we used for training and inference in TIPO.

## F IMAGE EXAMPLES

In this section, we present sample images from the experiments described in Section 5 to visually demonstrate the improvements achieved by TIPO.

### F.1 IN-DOMAIN TEST REGARD TO SCENERY TAG

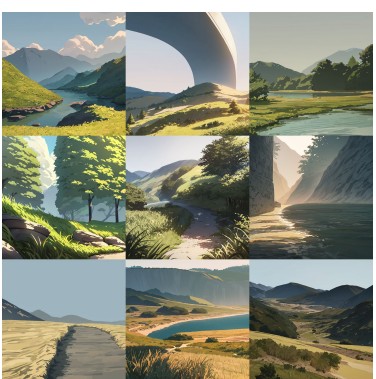 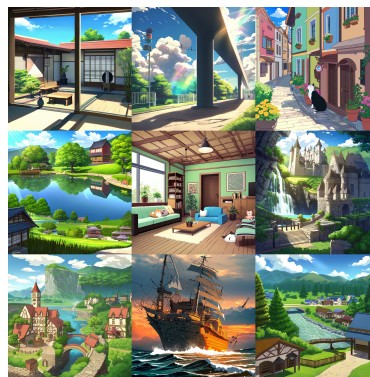

Figure 17: Comparison of generated images using simple input (left) vs. TIPO-enhanced input (right) for the scenery tag

Figure 17 demonstrates the difference in output diversity between simple input and TIPO-enhanced input for the scenery tag. As observed, TIPO significantly expands the range of generated sceneries, better reflecting the variety present in the Danbooru2023 dataset (Yeh, 2024b). The left column shows results from simple input (scenery tag only), while the right column illustrates the enhanced diversity achieved with TIPO-enhanced input.

### F.2 IN-DOMAIN PROMPT GENERATION TEST

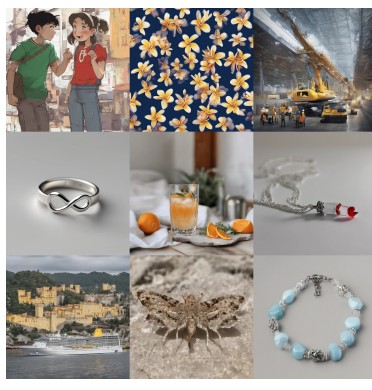 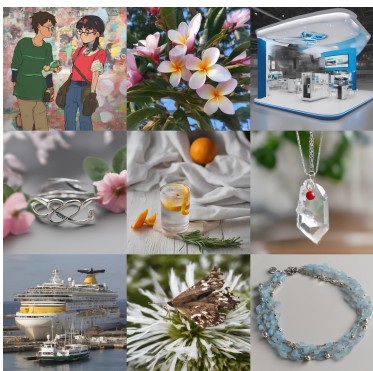

(a) Short Caption            (b) TIPO-Generated Caption

Figure 18: Comparison of generated images using original input (left) vs. TIPO-enhanced input (right)

Figure 18 illustrates the differences between short captions, truncated long captions, TIPO-generated captions, and TIPO-extended captions. The "short prompt" and "truncated long prompt" used in this experiment typically consist of 1-2 sentences, resulting in reasonably good quality outputs. However, the use of TIPO to refine or extend these prompts still yields noticeable improvements in aesthetics and overall quality.

## G  HUMAN PREFERENCE

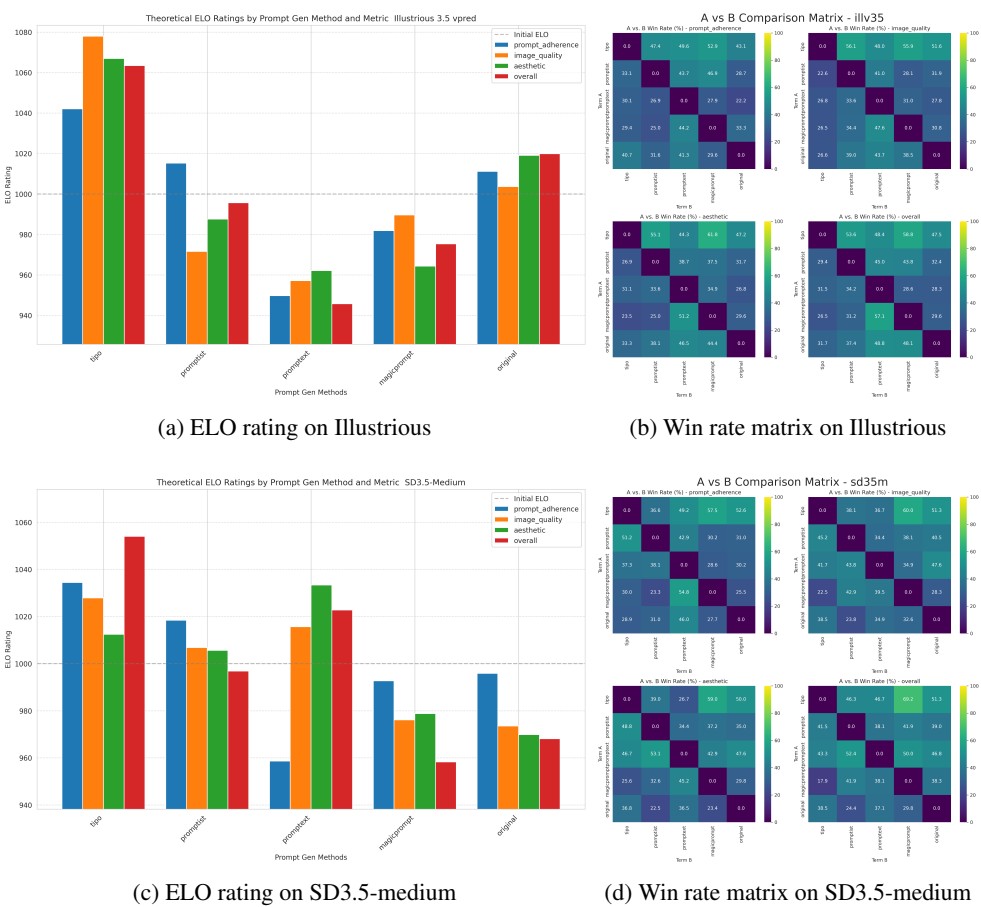

(a) ELO rating on Illustrious

(b) Win rate matrix on Illustrious

(c) ELO rating on SD3.5-medium

(d) Win rate matrix on SD3.5-medium

Figure 19: ELO ratings and win rate matrices across different experimental settings comparing five prompting methods (TIPO, Promptist, Promptextend, MagicPrompt, and Original) on three evaluation dimensions

| Comparison | Win Ratio (A:B) | Proportion A | $p$-value | Significant? |
|---|---|---|---|---|
| Original vs. PromptExtend | 45:66 | 0.405 | 0.0572 | Marginally |
| Original vs. Tipo | 28:80 | 0.259 | $< 0.0001$ | Yes*** |
| Original vs. Promptist | 35:64 | 0.354 | 0.0046 | Yes** |
| PromptExtend vs. Promptist | 41:80 | 0.339 | 0.0005 | Yes*** |
| Promptist vs. Tipo | 30:101 | 0.229 | $< 0.0001$ | Yes*** |
| PromptExtend vs. Tipo | 30:90 | 0.250 | $< 0.0001$ | Yes*** |
| MagicPrompt vs. Promptist | 14:34 | 0.292 | 0.0055 | Yes** |
| MagicPrompt vs. Tipo | 11:48 | 0.186 | $< 0.0001$ | Yes*** |
| MagicPrompt vs. Original | 15:28 | 0.349 | 0.0660 | No |
| MagicPrompt vs. PromptExtend | 19:35 | 0.352 | 0.0402 | Yes* |

Table 11: Pairwise win rates and statistical significance (Overall Dimension). *Significance levels: * $p < 0.05$, ** $p < 0.01$, *** $p < 0.001$*

We conducted a series of A/B tests to compare five prompt transformations, (*TIPO*, *Promptist*, *Promptext*, *MagicPrompt*, and *Original(unmodified)*), for two models, *Illustrious*, *SD3.5-medium*, which is known for both core word/natural language understanding. In total, we collected responses for ∼1,500 pairwise comparisons, from more than 20 anonymous evaluators. Each evaluation asked

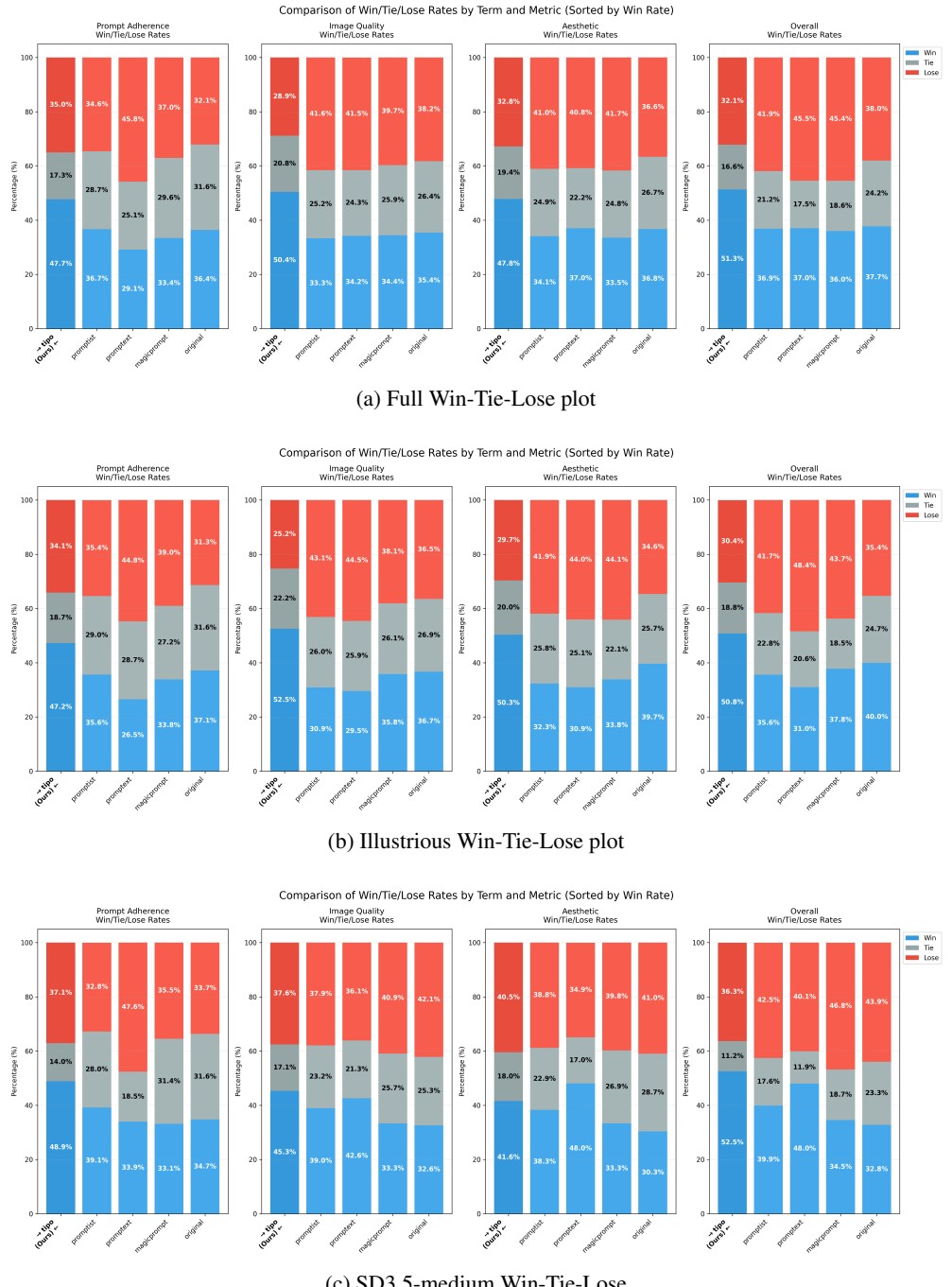

Figure 20: Win-Tie-Lose comparison across different experimental settings showing the relative performance of five prompting methods on prompt adherence, image quality, and aesthetic appeal metrics

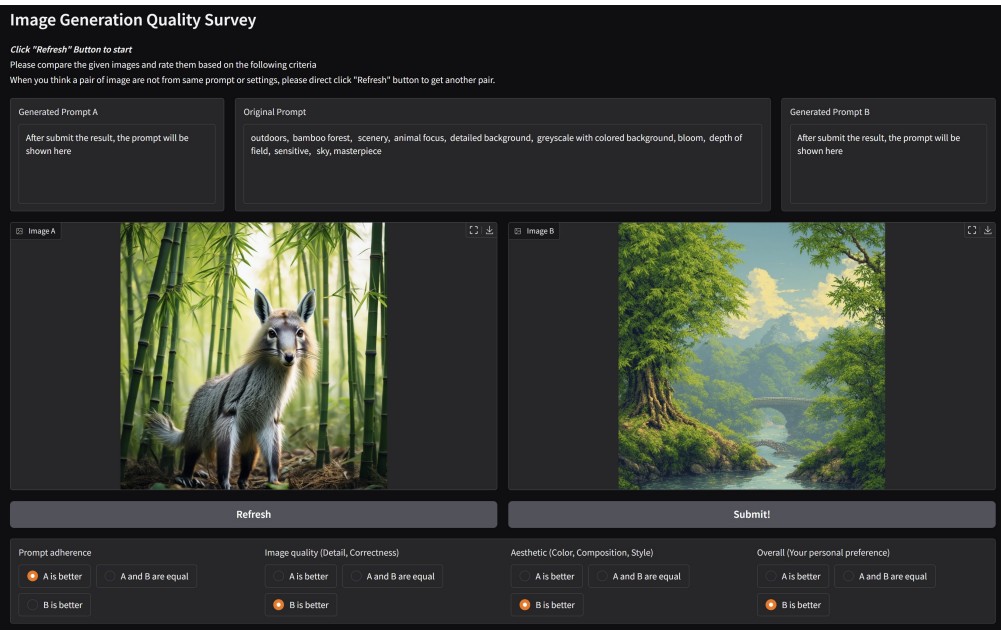

(a) The UI of survey system before submitting the choices.

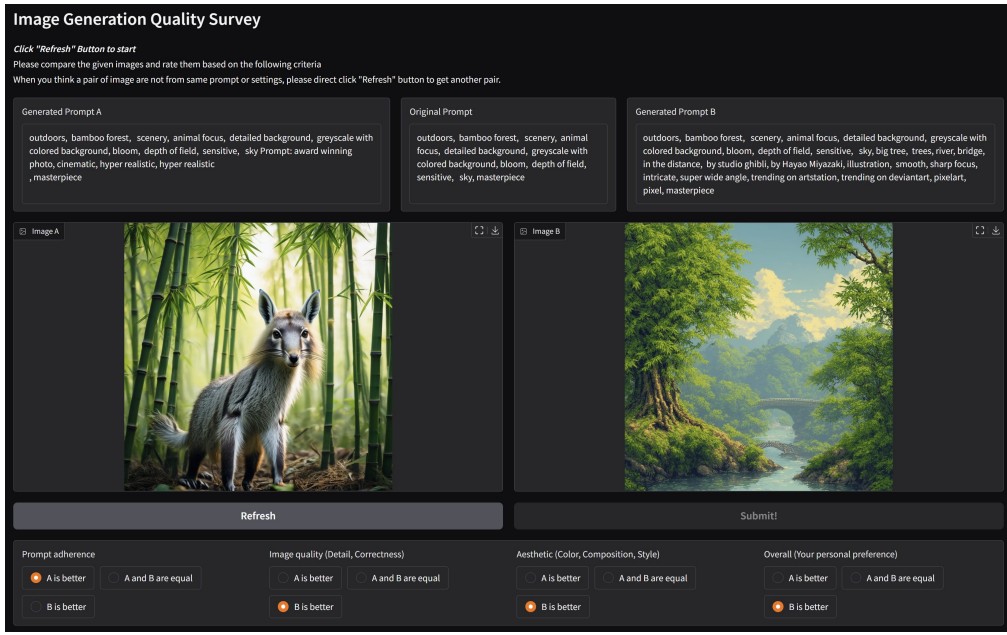

(b) The UI of survey system after submitting the choices.

Figure 21: Survey interface for human evaluation of image pairs, showing the evaluation process before submission (a) where users compare two images based on four metrics, and after submission (b) where the generated prompts for each image are revealed

participants to compare two generated images—labeled *A* and *B*—and select which they preferred (or a tie) according to specific criteria (e.g., prompt adherence, image quality, or aesthetic appeal).

## G.1 USER INTERFACE FOR HUMAN PREFERENCE EVALUATION

We developed a specialized survey interface to facilitate efficient and unbiased human evaluation of generated images. As illustrated in Figure 21, the interface presents evaluators with an original prompt and two corresponding images (labeled A and B) generated using different prompting methods. Before submission, users can see the original prompt in the center panel while the processed prompts used to generate each image remain hidden to prevent bias.

The evaluation framework requires participants to compare the image pairs across four distinct metrics: prompt adherence (how well the image follows the original prompt), image quality (detail and correctness), aesthetic appeal (color, composition, and style), and overall personal preference. For each metric, users can select one of three options: "A is better," "A and B are equal," or "B is better."

When evaluators encounter image pairs that appear to be from different prompts or settings, they are instructed to click "Refresh" to obtain a new comparison. After submitting their evaluations, the interface reveals the transformed prompts used to generate each image, providing transparency about how the original prompt was modified by each method.

## G.2 EXTENDED HUMAN EVALUATION.

Participants assessed each image's performance on prompt adherence, image quality, and aesthetic appeal, with visually shown unmodified and image pairs. TIPO exhibited superior outcomes in all comparison settings. Notably, it attained a 64.4% peak win rate (against MagicPrompt) under the Full scenario and 57.5% (also against MagicPrompt) under SD35-medium, emphasizing TIPO's proficiency in generating images that closely follow prompt specifications while maintaining visual coherence.

## G.3 ELO RATINGS.

We computed theoretical ELO ratings from the aggregated pairwise comparisons to quantify overall performance differences among the five methods. The rating update rules were based on each pair's binary outcome (win or lose), ignoring tie cases. The result is depicted in Figure 19, TIPO has secured the highest ELO rating over other models.

## G.4 HUMAN PREFERENCE ELO METHOD

We computed theoretical ELO ratings from human-judged pairwise preference data to quantitatively evaluate the relative performance of each prompting method. The ELO rating system, initially designed for ranking chess players, aggregates binary outcomes into numerical ratings representing comparative performance.

**Pairwise Outcomes.** Human evaluators assessed comparisons between methods, resulting in one of three outcomes:

- Method $i$ wins: assigned a score of 1 for method $i$, and 0 for method $j$.
- Method $j$ wins: assigned score 1 for method $j$, and 0 for method $i$.
- Tie: assigned score 0.5 to both methods.

**Conversion to ELO Differences.** Win and tie rates were converted to ELO rating differences using:

$$\text{Adjusted Win Rate} = \text{Win Rate} + \frac{\text{Tie Rate}}{2}$$

$$\text{ELO Difference} = 400 \times \log_{10}\left(\frac{\text{Adjusted Win Rate}}{1 - \text{Adjusted Win Rate}}\right)$$

To ensure numerical stability, extreme adjusted win rates were constrained as follows:

$$\text{ELO Difference} = \begin{cases} -800, & \text{Adjusted Win Rate} \leq 0.001 \\ +800, & \text{Adjusted Win Rate} \geq 0.999 \end{cases}$$

**Calculating Method ELO Ratings.** Final ELO ratings were determined by averaging each method's pairwise ELO differences and centering these averages around a baseline rating (e.g., 1000):

$$\text{ELO}_{\text{method}_i} = \text{Base Rating} + (\text{Average ELO Difference for method } i - \text{Overall Mean ELO Difference})$$

**Interpretation of ELO Scores.** Methods with higher ELO scores consistently outperform lower-scored methods. A rating difference of 400 points corresponds to a 90% expected win probability for the superior method.

### G.5 STATISTICAL SIGNIFICANCE

As summarized in Table 11, we conducted two-sided binomial and McNemar's tests ($p <0.05$) to assess the statistical significance of observed differences. The result confirms that TIPO's advantages are unlikely to be explained by random variation, which also supports a consistent performance hierarchy: TIPO ranks highest, followed by Promptist, PromptExtend, Original, and MagicPrompt. Collectively, these findings illustrate TIPO's robust, model-agnostic effectiveness and underscore the model-sensitivity of alternative methods, particularly Promptext and MagicPrompt.

## G.6 SURVEY RESPONSE EXAMPLES

In this section we provided some responses of our human preference survey as reference.

best quality, absurdres, indoors, flower shop, animal focus, detailed background, colorful background, general
colorful background, general, dark, sky, night, anaglyph, masterpiece
masterpiece

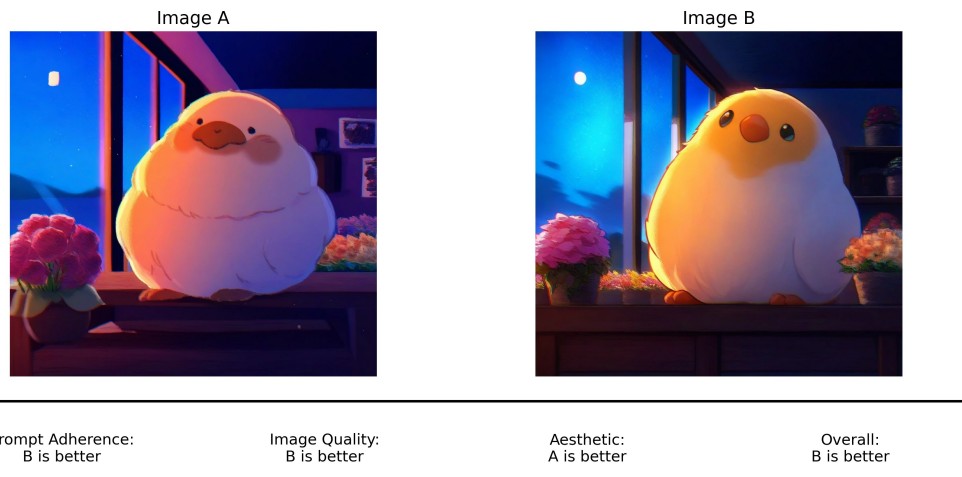

Prompt Adherence:     Image Quality:     Aesthetic:     Overall:
B is better     B is better     A is better     B is better

outdoors, animal focus, detailed background, gradient backgroundshade, shadow, darkness, bloom, depth of field
bloom, depth of field, sensitive, dark, cafe, anaglyph, scenery, depth of field
scenery, depth of field, masterpiece

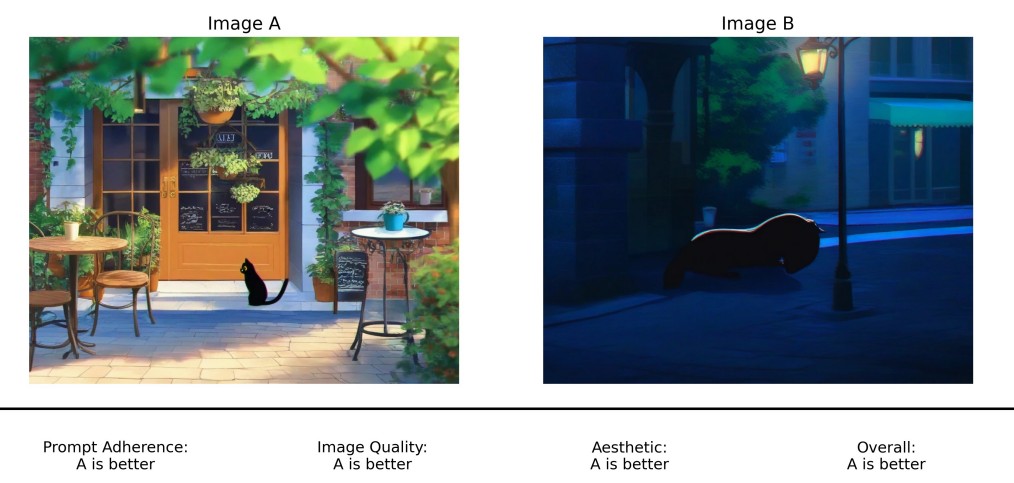

Prompt Adherence:     Image Quality:     Aesthetic:     Overall:
A is better     A is better     A is better     A is better

Figure 22: Some survey responses on illustrious-3.5-vpred generated image with different prompt optimization method

semirealistic, 1girl, high-waist skirt, headphones, detailed background, rainbow background, bloom, depth of field
bloom, depth of field, sensitive, fishes, transparent, scenery, backlighting, masterpiece
backlighting, masterpiece

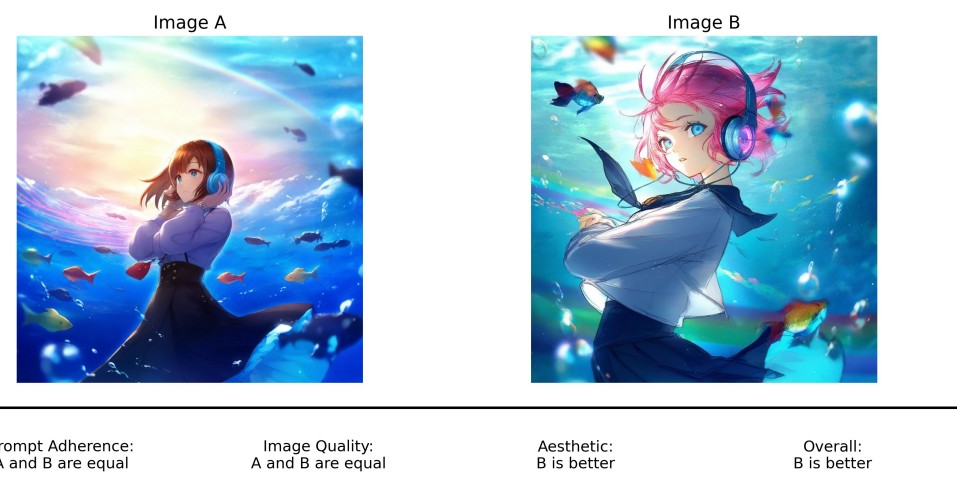

| Image A | Image B |

| Prompt Adherence: | Image Quality: | Aesthetic: | Overall: |
| A and B are equal | A and B are equal | B is better | B is better |

house, entrance, 1girl, Top Hat, bow, detailed background, rainbow background, sensitive
rainbow background, sensitive, classroom, fading, masterpiece

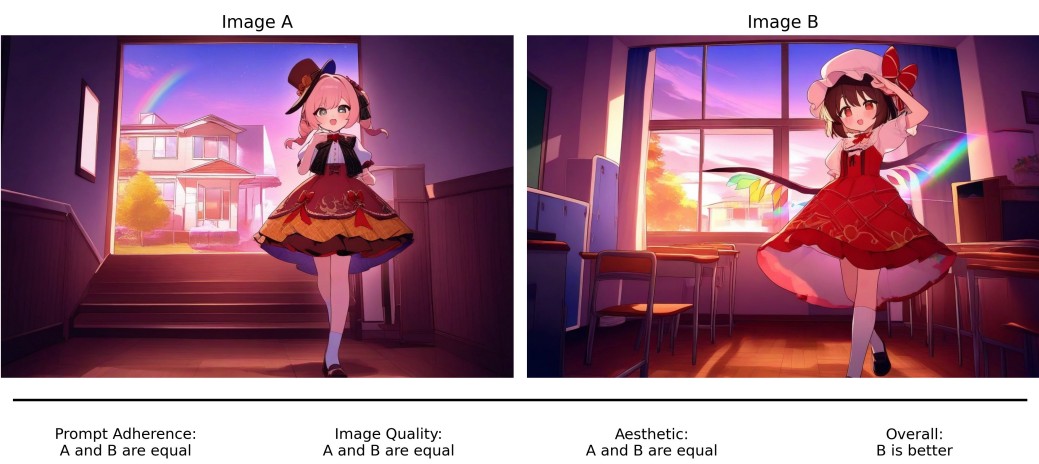

| Image A | Image B |

| Prompt Adherence: | Image Quality: | Aesthetic: | Overall: |
| A and B are equal | A and B are equal | A and B are equal | B is better |

Figure 23: Some survey responses on illustrious-3.5-vpred generated image with different prompt optimization method

semirealistic, scenery, animal focus, detailed background, two-tone backgroundshade, shadow, darkness, bloom
darkness, bloom, depth of field, general, sketch, scratch art, coffee, fading
coffee, fading, masterpiece

|              Image A              |              Image B              |
| :-------------------------------: | :-------------------------------: |

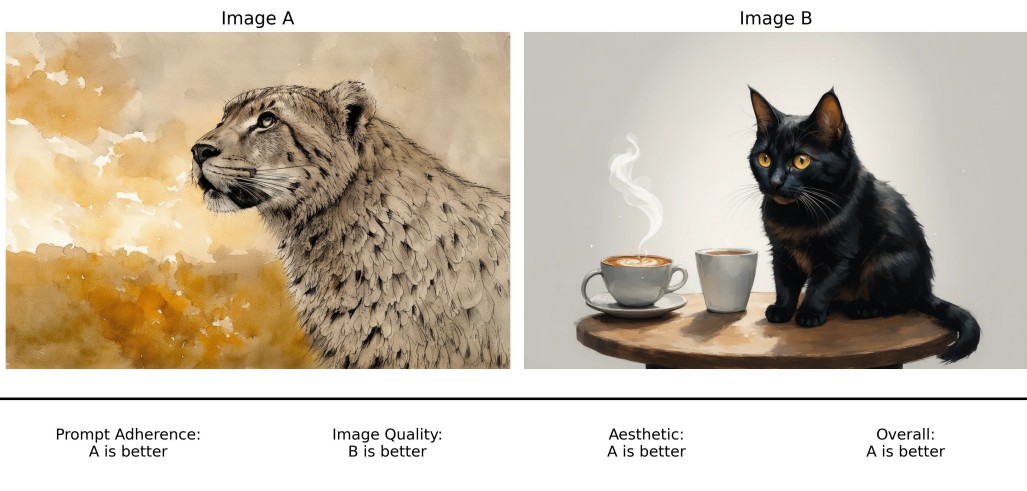

Prompt Adherence:          Image Quality:          Aesthetic:          Overall:
A is better                B is better             A is better         A is better

shrine, scenery, animal focus, detailed background, rainbow background, sensitive, sketch, scratch art
sketch, scratch art, funeral, fading, scenery, muted colors, greyscale, masterpiece
greyscale, masterpiece

|              Image A              |              Image B              |
| :-------------------------------: | :-------------------------------: |

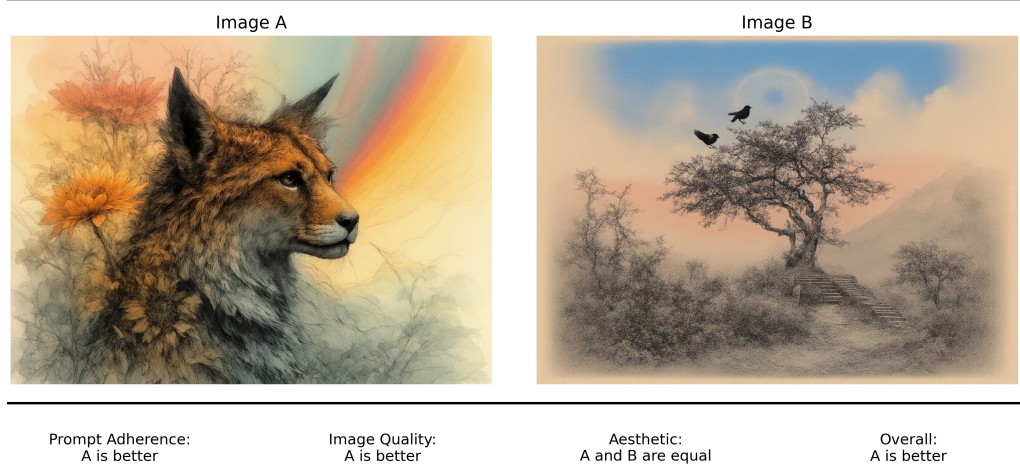

Prompt Adherence:          Image Quality:          Aesthetic:              Overall:
A is better                A is better             A and B are equal       A is better

Figure 24: Some survey responses on SD3.5-medium generated image with different prompt optimization method

semirealistic, 1girl, Jodhpurs, crowns, detailed background, greyscale with colored background, general, sketch
general, sketch, scratch art, ocean, fading, masterpiece

|  Image A  |  Image B  |
| --- | --- |

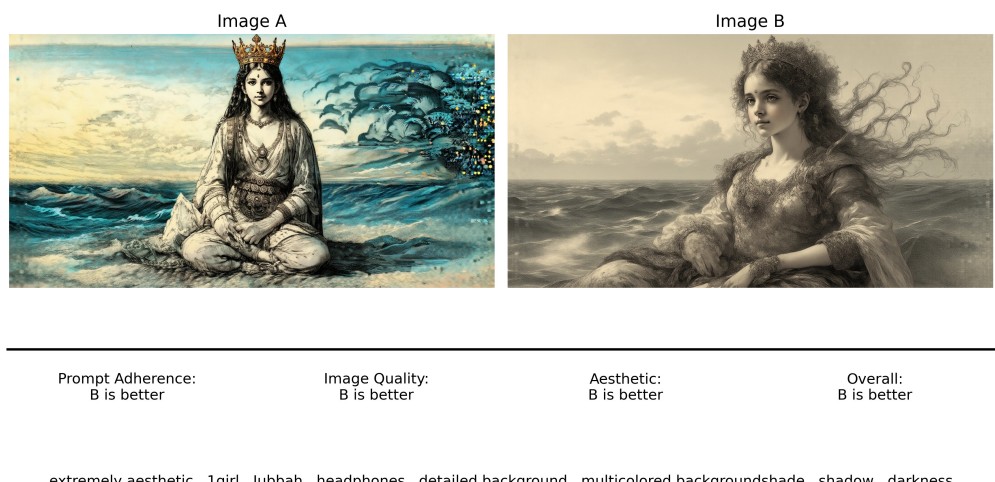

| Prompt Adherence: | Image Quality: | Aesthetic: | Overall: |
| B is better | B is better | B is better | B is better |

extremely aesthetic, 1girl, Jubbah, headphones, detailed background, multicolored backgroundshade, shadow, darkness
shadow, darkness, general, sketch, scratch art, coffee, transparent, scenery
transparent, scenery, JPEG artifacts on purpose, masterpiece

|  Image A  |  Image B  |
| --- | --- |

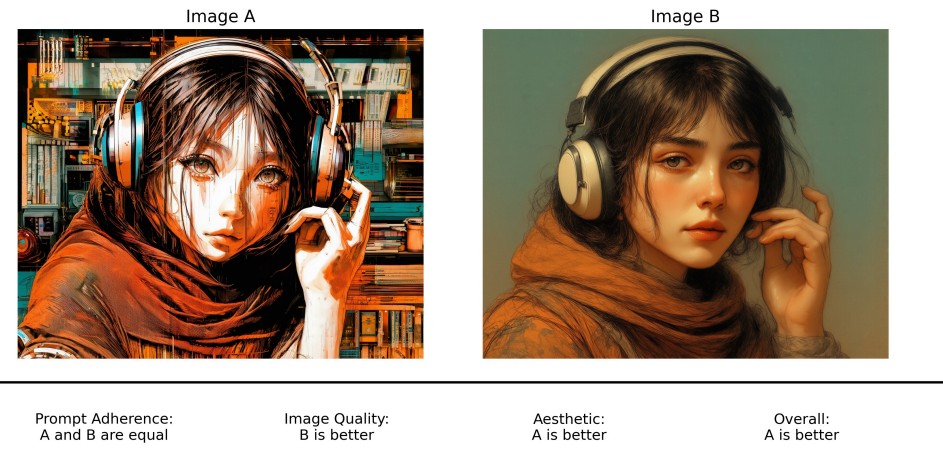

| Prompt Adherence: | Image Quality: | Aesthetic: | Overall: |
| A and B are equal | B is better | A is better | A is better |

Figure 25: Some survey responses on SD3.5-medium generated image with different prompt optimization method

## G.7   CONCLUSION

The extended evaluations presented here reinforce TIPO's standing as a reliable and effective prompt-optimization strategy. Its consistent performance gains under diverse model conditions highlight its potential for broad applicability, with its strong alignment with user-specified prompts, high image quality, and favorable aesthetic outcomes.

# H   ABLATION STUDY ON TIPO

In this section, we investigate the effect of incorporating **TIPO** (Tags + Inferred Prompt Objects) across various generation settings. Our primary goal is to validate whether additional structured information (e.g., core tags and minimal spatial/contextual cues) can improve image quality, reduce artifacts.

## H.1   EXPERIMENTAL SETUP

**Prompt Variants**   To systematically analyze TIPO's contribution, we consider four types of input prompts improvement task:

1. *Tag → More core words:* Given an initial set of core words, generate more refined or expanded core words.
2. *NL → More NL:* Given a short natural language (NL) description, elaborate into a richer NL prompt.
3. *Tag → (More core words + NL):* Combine expanded tags with a corresponding NL description derived from them.
4. *NL → (More NL + core words):* Use the NL prompt to add relevant tags, forming a mixed prompt of NL plus core words.

In each case, we compare the baseline prompts (without TIPO cues) against prompts incorporating TIPO's structured, tag-based critical information and minimal spatial hints.

**Data Preparation**   We start by randomly sampling core words from a word table to represent a diverse range of topics (e.g., objects, environments, descriptors). Additionally, for each word set, we generate a corresponding short NL sentence using a compact language model (GPT4o-mini). Overall, the six prompt variants are tested on 4,000 images, ensuring a balanced comparison.

**Inference Procedure**   Prompts are fed into our image-generation pipeline under identical model settings (classifier free guidance, sampler, steps, etc.), using the v-parameterized variant of *Illustrious v3.5*(Park et al., 2024). We focus on how TIPO modifications alter the generation outcomes and whether they introduce additional computational overhead.

## H.2   EVALUATION METRICS

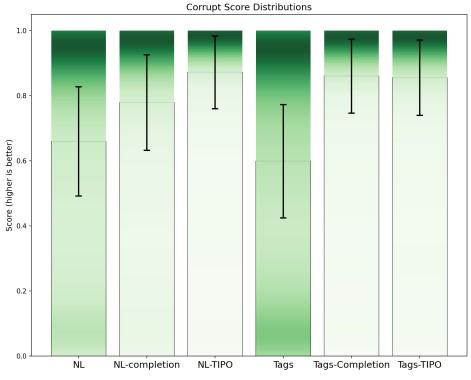

(a) Corrupt Score Distributions

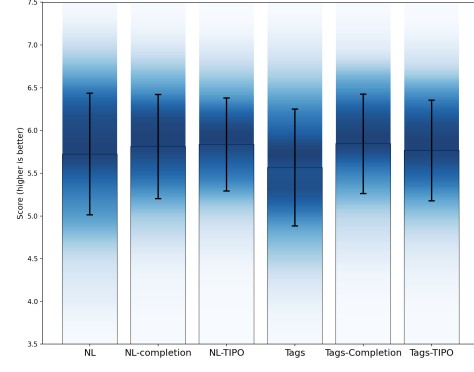

(b) Aesthetic Score Distributions

Figure 26: Side-by-side comparison of Corrupt (left) and Aesthetic (right) score distributions across prompt types.

**Aesthetic Score**   We employ an off-the-shelf aesthetic predictor to estimate image quality. In the following paragraph, we discuss the model's bias.

**AI Corruption Score**    Using an automated ' AI corruption ' detection model, we measure generation artifacts, such as distorted objects and unnatural shapes. Higher scores imply cleaner, more coherent outputs.

### H.3    RESULTS & DISCUSSION

**Impact on Aesthetics**    Figure 26b shows that **TIPO**-enhanced prompts generally achieve higher aesthetic scores than their non-TIPO counterparts, albeit with some variance. Notably, we observe a correlation between wider color ranges and higher aesthetic scores discussed in Figure 27, suggesting a bias toward more colorful or varied compositions.

**Improvements in Corruption Score**    As shown in Figure 26a, TIPO-based prompts yield significantly lower corruption scores, indicating fewer artifacts. We hypothesize that the additional spatial and contextual details encoded via TIPO help the model place objects more consistently.

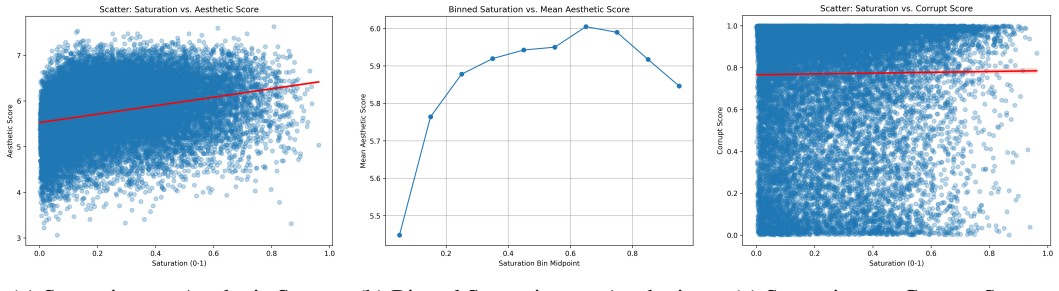

(a) Saturation vs. Aesthetic Score    (b) Binned Saturation vs. Aesthetic    (c) Saturation vs. Corrupt Score

Figure 27: Scatter plots (left and right) and binned analysis (middle) showing the relationship between saturation and image metrics. We find a moderate positive correlation between saturation and aesthetic score (Pearson $r = 0.2821$), particularly at lower saturation ranges, based on 24k samples. However, saturation shows no notable correlation with corrupt score (Pearson $r = 0.0125$).

### H.4    SPEED TEST AND OVERHEAD ANALYSIS

**Inference Speed**    A key concern for production pipelines is whether TIPO generation imposes a substantial time overhead. We benchmarked prompt-generation inference on four smaller models, excluding any large proprietary LLMs. As illustrated in Table 12, the additional TIPO-related computation remains well below the image-generation time. Hence, even in a synchronous pipeline, TIPO prompt expansion does not constitute a bottleneck.

Table 12: Speed Test Results for TIPO and Other Prompt Methods

| Method | Model/Config | # Runs | Avg. Time (s) | Std. Dev. (s) |
|---|---|---|---|---|
| | LLaMA-500M | 500 | 1.4207 | 1.0730 |
| TIPO | LLaMA-200M | 200 | 1.0306 | 0.8982 |
| | LLaMA-100M | 200 | 1.0078 | 0.9394 |
| PROMPTIST | GPT2-125M | 1000 | 1.4593 | 0.2857 |
| PROMPTEXTEND | GPT2-125M | 1000 | 1.3849 | 0.2151 |
| MAGICPROMPT | GPT2-125M | 1000 | 1.1398 | 0.4043 |

**Memory Footprint**    We also confirm that TIPO's overhead in terms of VRAM usage is minimal (e.g., $< 0.5$ GB for TIPO-200M and $< 1.5$ GB for TIPO-500M) with the quantization supported by llama.cpp. The practical adoption of TIPO in pipelines has shown no critical memory concerns, which aligns with our measurements.

**Training Costs**   To contextualize the training cost of TIPO, we compare its training time with reinforcement learning (RL)-based prompt optimization methods, based on the reported settings in their original papers. While it is technically difficult to reproduce RL-based methods on our hardware (4×RTX 3090), their reported GPU-hour budgets provide an approximate reference for scale and efficiency. The comparison is summarized in Table 13.

Although TIPO requires more total GPU-hours than PAE, it is trained on over 30 million prompts—two orders of magnitude larger than both Promptist and PAE. After normalization, TIPO achieves the lowest cost per 1k prompts, demonstrating strong scalability. It is also worth noting that Promptist and PAE rely on reinforcement learning with external T2I rollouts. Even for SD1.5, each rollout takes roughly five seconds, and the cost increases dramatically for larger models such as SDXL, SD3, or Flux. By contrast, TIPO requires no rollouts, so its training cost scales linearly with corpus size and remains independent of the target T2I model.

Table 13: Training cost comparison between TIPO and RL-based prompt optimization methods. GPU-hours per 1k prompts are normalized for fairness.

| Method | #Params | #Prompts | #GPUs | GPU-h | GPU-h /1k |
|---|---|---|---|---|---|
| TIPO | 200M | 30,000k | 4×RTX 3090 | 1,680 | **0.056** |
| PAE | 125M | 450k | 1×A800 | 90 | 0.20 |
| Promptist | 125M | 90k | 4×V100 (SFT), 32×V100 (RL) | 63 | 0.70 |

### H.5   CONCLUSION OF ABLATION

Our experiments suggest that **TIPO** (1) consistently lowers AI corruption artifacts, (2) can boost aesthetic scores, and (3) remains computationally inexpensive. The improvements in metrics support the viability of TIPO prompts for real-world image-generation tasks. In short, a concise natural language prompt with core tag-based critical information appears to be an effective, suggested form for most use cases.

## I   TOPIC DISTRIBUTION VISUALIZATION

Latent Dirichlet Allocation (LDA) (Blei et al., 2003) is a generative probabilistic model for topic modeling (Jelodar et al., 2018), which assumes that each document is a mixture of topics, with each topic represented by a distribution over words. LDA uncovers hidden thematic structures by analyzing word co-occurrence patterns, while methods like TF-IDF and TextRank (Mihalcea & Tarau, 2004) enhance its ability to extract meaningful insights from large textual datasets. We implemented a multi-stage topic modeling and clustering methodology using LDA to extract varying numbers of topics (20, 30, 50, and 100) from the corpus. This approach focuses on identifying significant representative words while filtering out stop words and irrelevant terms to ensure meaningful topic classification.

We empirically assessed whether the resulting topics were sufficiently large and diverse by employing multi-level topic analysis. This iterative process mitigates potential challenges such as substantial topic overlap, which can diminish distinctiveness when extracting a large number of topics (Stevens et al., 2012).

To address the potential overlap and further assess the diversity and meaningfulness of the topics, we performed a secondary clustering (Zhao & Karypis, 2002). We grouped the initially extracted topics into five major clusters using k-means clustering. We evaluated the clustering performance by calculating the inertia (Sum of Squared Distances) (Hartigan et al., 1979), shown in Table 14, 15, and 16. Since the topics have already been filtered for meaningful content, a higher inertia value indicates greater diversity among the clusters, reflecting a broader range of valid and meaningful topics across the dataset. This two-tiered approach allows for a more nuanced analysis of topic diversity and ensures the robustness of the topic modeling against meaningless word groupings.

| Size | MagicPrompt | | GPT4o-mini | | Promptist | | TIPO | |
|------|-------|-------|-------|-------|-------|-------|-------|-------|
| | Run 1 | Run 2 | Run 1 | Run 2 | Run 1 | Run 2 | Run 1 | Run 2 |
| 20 | 170.78 | 137.94 | 1078.38 | 204.71 | 125.90 | 144.59 | 1037.44 | 271.77 |
| 30 | 461.79 | 417.74 | 1758.40 | 736.74 | 327.48 | 195.13 | 1323.88 | 512.07 |
| 50 | 829.68 | 730.73 | 861.15 | 1036.33 | 400.90 | 373.74 | 823.51 | 959.18 |
| 100 | 1656.74 | 1245.60 | 1987.63 | 1628.32 | 877.36 | 657.14 | 1622.79 | 1777.61 |

Table 14: Inertia for COYO-Dataset inference, higher is better

| Size | MagicPrompt | | GPT4o-mini | | Promptist | | TIPO | |
|------|-------|-------|-------|-------|-------|-------|-------|-------|
| | Run 1 | Run 2 | Run 1 | Run 2 | Run 1 | Run 2 | Run 1 | Run 2 |
| 20 | 184.53 | 352.16 | 244.79 | 246.15 | 125.47 | 198.94 | 278.56 | 211.98 |
| 30 | 452.01 | 566.74 | 505.56 | 441.34 | 204.28 | 328.70 | 372.07 | 471.28 |
| 50 | 571.77 | 895.47 | 1227.30 | 990.17 | 438.89 | 313.48 | 737.65 | 788.41 |
| 100 | 1291.60 | 1742.36 | 1675.41 | 1550.32 | 631.61 | 628.78 | 1573.47 | 1855.90 |

Table 15: Inertia for GBC-Dataset inference, higher is better

We attach a simple visualization of topics in scenery prompt generation, with topic n=100, cluster k=5.

| Size | MagicPrompt | GPT4o-mini | Promptist | TIPO |
|------|-------------|------------|-----------|---------|
| 20 | 60.82 | 734.52 | 210.60 | 139.29 |
| 30 | 275.76 | 1141.77 | 415.95 | 355.20 |
| 50 | 630.50 | 826.29 | 722.75 | 1002.36 |
| 100 | 2026.39 | 1879.08 | 802.93 | 1883.70 |

Table 16: Inertia for Scenery extend inference, higher is better

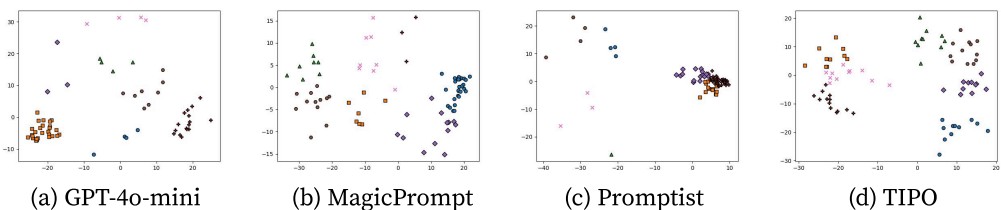

(a) GPT-4o-mini  (b) MagicPrompt  (c) Promptist  (d) TIPO

Figure 28: Topic visualization for scenery prompt generation. A wider spread indicates a greater diversity of generated topics.

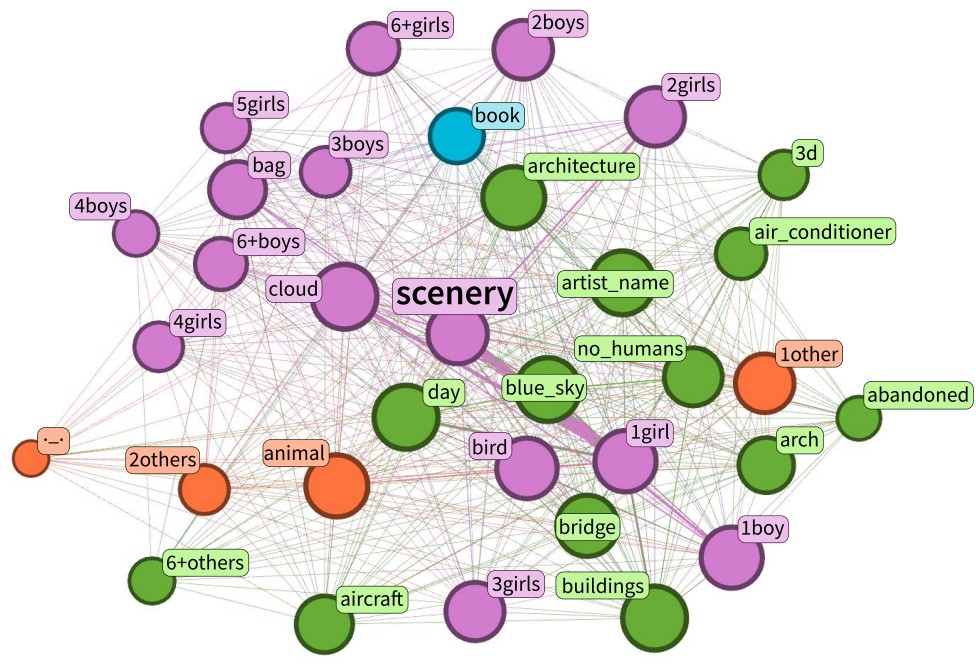

Figure 29: This visualization represents a filtered subset of posts from the Danbooru2023 dataset, centered on the 'scenery' tag. The network graph is an ego network (depth 1), which includes only nodes directly connected to the 'scenery' tag. To refine the data and focus on meaningful associations, uncommon tags with fewer than 10 occurrences were excluded. The analysis, conducted using Gephi, focuses on nodes with a degree greater than 600 to highlight critical components. Nodes are color-coded by modularity class by Fast Unfolding Algorithm (Blondel et al., 2008), revealing clusters of closely associated tags. Node size reflects Eigenvector Centrality(Bonacich, 1972), emphasizing highly connected and influential tags within their network.

## J  DISCUSSION AND FUTURE WORK

Despite the promising performance demonstrated by TIPO, several limitations and future directions remain open for further study.

**Distribution Dependence and OOD Generalization.**  The optimization behavior of TIPO inherently depends on the distributional bias of the open text–image corpus used for training. As a result, it may exhibit unstable behavior on extremely rare or stylistically unconventional prompts. Besides, when applied to out-of-distribution (OOD) T2I models whose training data or prompting conventions deviate significantly from LAION-style distributions, aesthetic performance may degrade slightly. In addition, our current OOD evaluation uses GPT-4o-mini–generated prompts, which are high-quality but do not fully represent real-world user queries, thus limiting external validity. Addressing the generalization to unseen models and long-tail prompts remains an important direction for future work, which can be pursued from two complementary perspectives: improving the model's generalization capacity via stronger backbones and domain-diverse fine-tuning, and enhancing evaluation through large-scale collection of authentic user prompts and cross-model benchmarking.

**Stronger Backbone Initialization.**  Our current implementation trains a mid-sized LLaMA variant from scratch. Future work could instead initialize from stronger open-source LLM backbones and fine-tune them on T2I corpora, potentially improving robustness on long-tail and domain-specific distributions. This could also mitigate failures on small or highly biased datasets by leveraging more general linguistic priors.

**Model-Specific Adaptation and Style Variance.**  For models that require structured or JSON-formatted prompts, TIPO can be combined with a lightweight adapter or fine-tuned on a small set of model-specific data. Extending TIPO with such adaptation modules could better accommodate systems whose prompt syntax diverges from mainstream diffusion models. In the longer term, TIPO can also serve as a backbone integrated with RL-based refinement for model-specific alignment.

**Personalization and Style Preservation.**  The current TIPO is a general-purpose optimizer and does not incorporate user-level or stylistic conditioning. Building on prior personalization techniques such as LoRA (Hu et al., 2022), future work could explore lightweight adapters or online learning mechanisms that track user preferences and maintain project-level stylistic consistency, enabling personalized and context-aware prompt optimization.

**Image-Aware and Feedback-Driven Refinement.**  TIPO currently operates purely in the text domain without utilizing generated images or user feedback. A promising extension is to incorporate vision–language models like Qwen3-VL (Bai et al., 2023) for image-aware refinement, allowing iterative refinement with optimized prompts, visual outcomes, and user instructions. However, such integration requires non-trivial data curation and training pipelines, which we leave for future work.

**Scaling Behavior and Model Capacity.**  From 200M to 500M parameters, TIPO continues to yield improvements in FDD and Aesthetic scores. Due to limited compute resources, we could not explore larger configurations to observe scaling saturation. A systematic study of TIPO's scaling behavior and architectural variants would be a valuable direction for future research. Such analysis could reveal scaling laws unique to prompt optimizers and guide practical model sizing for future deployments.

## K  DISCLOSURE OF LLM USAGE

We used GPT-5 only to polish writing by improving the readability and grammar correctness. No LLMs were used in the main contributions of this work, such as ideation, experiment design, or result analysis.

