# OpenReview forum: "TIPO: Text to Image with Text Pre-sampling for Prompt Optimization"
_ICLR.cc/2026/Conference — ICLR 2026 Poster_

### Official Review · Reviewer_TZN3 · 2025-10-29

**Soundness:** 2
**Presentation:** 3
**Contribution:** 2
**Rating:** 6
**Confidence:** 4

**Summary:**

TIPO (Text-to-Image Prompt Optimization) is an efficient framework designed for the automatic refinement of simple user inputs into rich, detailed prompts for text-to-image generation. The core intuition is to align the optimized prompt with the text distribution used in text-to-image model training. Compare with using expensive LLM rewriting or reinforcement learning approach, TIPO trains a lightweight multi-task language model, employing a pre-sampling mechanism to expand the original user prompt. This multi-task training includes pretext tasks for handling both tag-based and natural language inputs. Experiments demonstrate that TIPO achieves superior image quality, stronger text alignment and a 62.8% human preference win rate against strong baselines, while also providing up to a 29.4% runtime efficiency improvemen.

**Strengths:**

1. The core intuition is that optimal prompts must align with the large-scale text distributions used to train T2I models, which can reduce the mismatch between the training and inference. It is novel and straightforward. Moreover, the method is designed to be universal, leveraging its large-scale curated corpus of over 30 million text descriptions to ensure compatibility across various T2I models.

2. The method employs a lightweight language model to expand the use prompts. This computational efficiency is a major practical advantage over resource-intensive LLM-based or RL-based methods.

3. The experimental results are solid. It is conducted on both in-domain and out-of-domain tasks, which shows that the method is general.

**Weaknesses:**

1. The limitation of this work is that it needs the large-scale open-sourced text-to-image training data to make sure that prompt expanding is robust. But some current models are trained use higher-quality close-sourced data, often containing some synthetic data. It is impossible  to get the reasonable data distribution of these models, making the proposed method ineffective.

2. The framework does not consider any feedback from the generated images, which means that it cannot adaptively improve the prompts according to user feedback or model feedback.

**Questions:**

What sampling method do you use, top-p or greedy decoding?

---

> ### Author Response · Authors · 2025-11-15
>
> Thank you for your encouraging review. We sincerely appreciate your recognition of the distribution-alignment intuition, the efficiency of the lightweight expansion model, and the generality demonstrated by our experiments. Your comments are motivating and helpful. We now address the concerns you raised.
>
> ## W1. On dependence on large-scale open-source training data
>
> We acknowledge that TIPO relies on large-scale open-source text–image corpora to obtain a robust distribution for prompt expansion. However, our experiments already show that this reliance does not prevent TIPO from being effective on current SOTA models with close-source training data. Among the five models evaluated (FLUX.1-dev, Omnigen2, Lumina-2, HiDream-I1, and Gemini-Flash-2.0-Image, all with close-source training data), four exhibit clear aesthetic improvements with TIPO-optimized prompts (Table 3). This suggests that, despite partial use of proprietary or synthetic data, these models still retain substantial overlap with the broader text distributions found in open-source corpora.
>
> For models whose prompting conventions differ dramatically from mainstream systems, we agree that TIPO may struggle. In such cases, TIPO may serve as a backbone and be combined with existing model-specific strategies (e.g., format adapters or RL-based tuning) to better match the target distribution. We will clarify this limitation in the revision and highlight model-specific adaptation as a direction for future work.
>
> ## W2. Lack of image-aware or feedback-driven refinement
>
> Thank you for this valuable observation. We agree that TIPO does not incorporate any feedback from generated images, and that enabling adaptive refinement based on user feedback or model feedback is an important direction for improving usability. After detailed analysis, we believe this will likely require finetuning instruction-following vision-language models (e.g., Qwen3-VL) to support image-conditioned prompt refinement, but the construction of large-scale datasets, task formulations, and training pipelines presents nontrivial technical challenges. We very much appreciate this suggestion and will highlight it as a promising direction for future work.
>
> ## Q. Sampling strategy
>
> Thank you for the question. We use a hybrid stochastic decoding strategy combining **top-p = 0.95** and **top-k = 60**, which follows standard practice in open-ended text generation, as used in the official HuggingFace generation examples [1] and in recent academic work [2]. We therefore follow this established default.
>
> **References**
>
> [1] Hugging Face. “Usage — transformers 2.11.0 documentation.” Example of text generation with XLNet using top_p = 0.95 and top_k = 60. https://huggingface.co/transformers/v2.11.0/usage.html
>
> [2] Faysse, M. et al. “CroissantLLM: A Truly Bilingual French–English Language Model.”  arXiv:2402.00786, 2024.

---

> > ### Author Response · Authors · 2025-11-22
> > **Manuscript Update**
> >
> > Thank you for your insightful comments. In the revised **Appendix J. Discussion and Future Work**:
> >
> > - We added a discussion on distribution dependence and out‑of‑distribution generalization, and outlined steps for better generalization.
> > - We highlighted potential extensions involving image‑aware and feedback‑driven refinement using vision–language models.
> >
> > Moreover,
> >
> > - We introduced our sampling strategy in **Appendix C.3 TIPO Inference Settings**, detailing a hybrid nucleus sampling (top‑p = 0.95) and top‑k = 60 decoding approach.
> >
> > Edits are highlighted in **blue** in the revised PDF.
> > Thank you for helping improve the paper’s scope and clarity.

---

### Official Review · Reviewer_LKuZ · 2025-10-31

**Soundness:** 2
**Presentation:** 2
**Contribution:** 2
**Rating:** 4
**Confidence:** 3

**Summary:**

The proposed TIPO is a prompt optimization method for text-to-image (T2I) models that leverages a lightweight pre-trained model to automatically expand users' simple inputs into detailed and semantically rich prompts. The core idea is to transform user input prompts into prompts that align with the text distribution of T2I model training data, thereby significantly improving visual quality, coherence, and detail while preserving the original intent. The experiments report that prompt refinement can be performed in a lightweight, fast, and general manner across various evaluation metrics such as FDD and Aesthetic Score.

**Strengths:**

- The implementation approach, which uses a LLaMA-based 200M parameter model for pre-training and does not depend on large-scale LLMs or reinforcement learning, is attractive from a practical standpoint. Additionally, it does not involve fine-tuning of T2I models.

- For tag-based prompts, TIPO achieves the best FDD (0.2282), and for natural language-based (short) and truncated long prompts, it demonstrates the best or second-best performance in Aesthetic Score and AI Corrupt Score with stable performance.

- Beyond Stable Diffusion family models, performance improvements are confirmed on diverse T2I models including FLUX.1-dev, Omnigen2, Lumina-2, and HiDream-I1. The fact that TIPO is effective even for models with undisclosed training data demonstrates the high generalizability and robustness of the proposed method.

- While conventional methods indirectly optimize prompts using reward models, the finding that better prompts can be created simply by directly aligning with the data distribution learned by T2I models, as in the proposed method, provides an interesting insight.

**Weaknesses:**

- The verification of "distribution-aligned" is indirect. While claiming to "align with T2I training distributions," the paper does not directly measure the distance of text distributions themselves (e.g., KL divergence of embedding distributions or perplexity differences). The alignment is primarily inferred from image-side metrics (such as FDD) and human comparisons, lacking direct measurement of text distribution.

- In OOD settings, while diversity (Vendi Score) improves significantly, there is an issue that GPT-generated prompts are accurate but lack diversity. Although TIPO optimization adds additional details that harmonize with the original themes and significantly improves the diversity of generated images, the relative positioning of aesthetic scores and corruption rates compared to some baselines should be explicitly stated as a limitation of generalizability.

- While the paper presents inference time cost comparisons (Table 4 showing up to 29.4% speedup), and Table 6 shows training settings for each TIPO model (GPU types, training time, datasets), there is insufficient direct quantitative comparison with RL-based methods such as Promptist and PAE regarding training costs. Explicit comparison is desired for fair evaluation.

- The adopted metrics are primarily image-centric, such as Aesthetic Score, AI Corrupt Score, Vendi Score, and FDD, and do not include automatic metrics that measure text-image alignment such as CLIPScore or GenEval. There is insufficient quantitative evidence to support the claimed "stronger text alignment."

**Questions:**

1. Can you add metrics that directly measure text distribution alignment (e.g., embedding distance from training data or perplexity)?

2. While diversity improves in OOD environments, do you have any causal analysis or improvement strategies regarding the relative performance differences with other baselines?

3. Can you provide a quantitative comparison of training computational costs and resource requirements (GPU memory, training time, etc.) with RL-based methods such as Promptist and PAE?

4. Can you add evaluation results using automatic metrics that directly measure text-image alignment, such as CLIPScore or GenEval?# Review of "TIPO: Text to Image with Text Presampling for Prompt Optimization"

---

> ### Author Response · Authors · 2025-11-15
>
> We sincerely thank you for the thoughtful and insightful comments. Your feedback greatly improved the conceptual clarity and empirical grounding of our work. **In particular, your comment on “distribution alignment” was especially meaningful for us.** We had not initially considered that our intuition of "good prompts should align with ... T2I training" could be directly validated. Your review reframed it as a testable hypothesis, motivating us to add quantitative analyses and provide clear empirical support. We are sincerely grateful for this insight.
>
>
> ## W1 and Q1. Direct Measurement of Text Distribution Alignment
>
> Following your suggestion, we directly measured embedding-space distances between optimized prompts and T2I training text corpora. We sampled 1,000 natural-language captions from COYO and 1,000 tag-based captions from Danbooru2023. We then encoded ground-truth captions and the outputs of all optimization methods using two widely adopted encoders: T5-XXL [1] (used in SD3, Flux, PixArt) and jina-embeddings-v3 [2] (SOTA in text embedding) . We computed Fréchet Distance (FD) and Maximum Mean Discrepancy (MMD, RBF kernel). KL divergence is not used because these embeddings do not form explicit probability distributions. Lower is better for all metrics.
>
> **Encoder: Jina-embeddings-v3**
>
> | Prompt Type | Metric | TIPO       | Promptist | MagicPrompt | GPT-4o-mini |
> | ----------- | ------ | ---------- | --------- | ----------- | ----------- |
> | NL-short    | FD     | **0.0322** | 0.1003    | 0.0385      | 0.1064      |
> |             | MMD    | **0.0320** | 0.1501    | 0.0624      | 0.1699      |
> | NL-trunc    | FD     | **0.0309** | 0.1192    | 0.0493      | 0.0963      |
> |             | MMD    | **0.0359** | 0.1700    | 0.0772      | 0.1642      |
> | Tag-based   | FD     | **0.1094** | 0.1891    | 0.1958      | 0.2479      |
> |             | MMD    | **0.1539** | 0.2473    | 0.2415      | 0.3050      |
>
> **Encoder: T5-XXL**
>
> | Prompt Type | Metric | TIPO       | Promptist | MagicPrompt | GPT-4o-mini |
> | ----------- | ------ | ---------- | --------- | ----------- | ----------- |
> | NL-short    | FD     | **0.0704** | 0.2072    | 0.1441      | 0.1252      |
> |             | MMD    | **0.1438** | 0.2914    | 0.1972      | 0.2297      |
> | NL-trunc    | FD     | **0.0674** | 0.2312    | 0.1884      | 0.1276      |
> |             | MMD    | **0.1404** | 0.3147    | 0.2270      | 0.2323      |
> | Tag-based   | FD     | **0.0524** | 0.2080    | 0.2578      | 0.0728      |
> |             | MMD    | **0.1846** | 0.3573    | 0.3948      | 0.2194      |
>
> Across all datasets, prompt types and encoders, TIPO achieves the lowest distance to T2I training text distributions, directly supporting our distribution alignment intuition. Thank you again for such a constructive suggestion.
>
> ## W2 and Q2. OOD Analysis and Limitations
>
> Thank you for the incisive concern regarding the OOD results. As you correctly pointed out, when the target T2I model is trained on a text distribution far from mainstream datasets such as LAION, refined prompts from TIPO may show a noticeable drop in aesthetic quality. This occurs because TIPO cannot fully align with aesthetic finetuning corpora from very different domains such as SD3.5 large. **We agree that this should be stated clearly as a limitation of generalizability.**
>
> For further analysis, we note that prompt optimization on black-box OOD T2I models is intrinsically difficult and outside the scope of this work. Moreover, our OOD experiments are intrinsically limited by the use of GPT-4o-mini generated prompts as a substitute for real user queries. These prompts concentrate on well-established artistic themes and are already coherent, polished and internally harmonious. Any further transformation, whether from Promptist, MagicPrompt or TIPO, tends to disturb this coherence, leading to lower aesthetic metrics and additional artifacts. Even so, TIPO introduces fewer artifacts than the baselines and achieves the strongest diversity improvements.
>
> While we plan to collect real user prompts for more rigorous OOD evaluation and causal analysis, this is not feasible during the discussion period due to time, cost, and privacy constraints. We sincerely hope you can understand these difficulties.

---

> ### Author Response · Authors · 2025-11-15
>
> ## W3 and Q3. Training Cost Comparison with RL-based Methods
>
> We agree that explicit comparisons improve clarity. While it is technically difficult to train RL-based methods on our hardware (RTX 3090), we summarize the reported costs from the official papers below.
>
> | Method    | #Params | #Prompts | #GPUs                          | Wall-clock time | GPU-hours | GPU-hours per 1k prompts |
> | --------- | ------- | -------- | ------------------------------ | --------------- | --------- | ------------------------ |
> | TIPO      | 200M    | 30,000k  | 4 × RTX 3090                   | 150 h + 270 h   | 1,680     | **≈ 0.056**              |
> | Promptist | 125M    | 90k      | 4 × V100 (SFT), 32 × V100 (RL) | Unknown         | 63        | ≈ 0.70                   |
> | PAE       | 125M    | 450k     | 1 × A800                       | 18 h + 3 days   | 90        | ≈ 0.20                   |
>
> - These GPU hours are not directly comparable because TIPO has more parameters and uses consumer GPUs, while Promptist and PAE rely on V100 and A800 clusters with much higher throughput.
>
> - Although TIPO uses more total GPU hours than PAE, it is trained on more than 30 million prompts, two orders of magnitude more than Promptist and PAE. After normalization, TIPO achieves the lowest cost per 1k prompts, indicating high scalability.
>
> - Promptist and PAE rely on RL with external T2I rollouts. Even for SD1.5, each rollout takes about five seconds and for larger models such as SDXL, SD3 or Flux, the cost grows dramatically. TIPO requires no rollouts, so its training cost scales linearly with corpus size and is independent of the target T2I model.
>
> We will highlight these results in the revised manuscript.
>
> ## W4 and Q4. Direct Text-Image Alignment Evaluation
>
> Following your suggestion, we conduct CLIPScore evaluations using openai/clip-vit-large-patch14-336 with the same input prompts as in W1. We do not evaluate GenEval [3] because its required prompting format (i.e., structured templates specifying explicit objects/relations) does not align with the more open-ended prompt formats used by our tested methods.
>
> | Prompt Type | TIPO          | MagicPrompt   | GPT-4o-mini | Promptist |
> | ----------- | ------------- | ------------- | ----------- | --------- |
> | Tag-based   | **0.2217**    | 0.1782 | 0.1774      | 0.1642    |
> | NL-short    | 0.2413 | **0.2834**    | 0.2378      | 0.2347    |
> | NL-trunc    | 0.2310 | **0.2517**    | 0.2275      | 0.2063    |
>
> Overall, TIPO achieves the strongest alignment for tag-based prompts and the second best on natural language prompts. MagicPrompt’s higher score on natural language prompts stems from its 1.8M SD1.5 community training corpus, where users tend to write highly explicit and stylistically strong prompts. These match particularly well with the SD1.5 text encoder (also CLIP) and therefore naturally produce high alignment scores. However, such strong stylistic bias is often not preferred by real users, consistent with our human preference evaluation where MagicPrompt shows the lowest win rate (As shown in Table 9, where MagicPrompt vs TIPO win-ratio=11:48).
>
> ## Summary
>
> Thank you again for helping us strengthen the paper in several important ways. Your comments led us to validate our distribution alignment intuition with direct measurements, clarify the limitations of the OOD setting, provide a fairer training cost comparison and include explicit text image alignment metrics. We genuinely appreciate the level of insight in your review.
>
> [1] Raffel, C. et al. Exploring the Limits of Transfer Learning with a Unified Text-to-Text Transformer. JMLR, 2020.
>
> [2] Sturua, S. et al. jina-embeddings-v3: Multilingual Embeddings With Task LoRA. arXiv:2409.10173, 2024.
>
> [3] Ghosh, D., Hajishirzi, H. and Schmidt, L. GenEval: An Object-Focused Framework for Evaluating Text-to-Image Alignment. NeurIPS 2023.

---

> > ### Author Response · Authors · 2025-11-22
> > **Manuscript Update**
> >
> > Thank you again for your constructive suggestions. We have made several key additions:
> >
> > - **Prompt distribution alignment**: We added a new experiement compares embedding distances (FD & MMD) between optimized prompts and COYO/Danbooru captions using T5‑XXL and jina‑embeddings; **Table 5** demonstrates TIPO’s superior alignment.
> > - **Prompt–image alignment**: We added CLIPScore evaluation between prompts and generated images; **Table 6** shows that TIPO maintains strong alignment.
> > - **Training cost comparison**: Appendix H.5 includes **Table 13**, comparing GPU‑hours with RL‑based methods and showing that TIPO has the lowest normalized cost.
> > - **Discussion of distribution dependence**: Appendix J now explicitly discusses the limitations of distribution dependence and OOD generalization.
> >
> > Edits are highlighted in **blue** in the revised PDF.
> > We appreciate your insights; they strengthened the analysis and discussion.

---

> > > ### Comment · Reviewer_LKuZ · 2025-11-27
> > >
> > > Thank you for your detailed and sincere response. I acknowledge that most of the concerns I initially raised have been adequately addressed, and the quality of the paper has significantly improved. In particular, the direct measurement of text distribution alignment, the quantification of training cost comparisons, and the addition of CLIPScore-based alignment evaluation provide strong empirical support for the claims made in this work.
> > >
> > > However, the fundamental constraint of dependence on training data distribution remains a challenge, particularly when applying the method to emerging models trained on proprietary datasets. While this limitation should be explicitly stated, it does not undermine the core value of the proposed method. Taking all factors into consideration, I am raising my score.

---

> > > > ### Author Response · Authors · 2025-11-27
> > > >
> > > > Thank you again for your thoughtful engagement and for helping us improve the quality of the paper. Your incisive observations and constructive suggestions were invaluable, and we truly appreciate the time and care you invested in this review.

---

### Official Review · Reviewer_TMB7 · 2025-10-31

**Soundness:** 3
**Presentation:** 3
**Contribution:** 3
**Rating:** 6
**Confidence:** 3

**Summary:**

TIPO (Text-to-Image Prompt Optimization) proposes an efficient approach to automatic prompt enhancement for T2I models. Instead of fully rewriting user inputs with a large LLM, TIPO uses a lightweight, multi-task pretrained language model to perform text presampling before image generation: it retains the original user prompt and appends a structured, detail-rich continuation that better matches the text distribution T2I models are actually trained on.

**Strengths:**

The paper’s key strength is conceptual clarity: it treats “bringing the prompt back to the T2I training text distribution” as the central objective. To achieve this, it adopts a lightweight model that does prefix-preserving, suffix-expansion prompt optimization, which (i) avoids the off-topic drift often seen in full LLM rewrites, and (ii) is cheaper, faster, and more deployable than RL-based prompt optimization. The experimental section is also solid: it covers in-domain and out-of-domain settings, multiple T2I backbones, and evaluates with FDD, Aesthetic Score, AI Corrupt Score, Vendi, plus human preference, so the claimed gains are reasonably well supported.

**Weaknesses:**

The method is still distribution-dependent: if the user prompt is very niche, domain-specific, or stylistically unusual, the expanded prompt may not be reliable. Once the prompt is made more detailed, the generation space narrows, and Vendi indeed drops in some settings. For closed-source or stylistically distant T2I models, the “distribution-aligned” expansion can sometimes misfire, leading to slight aesthetic regressions. Finally, the current TIPO is a generic optimizer — it doesn’t condition on user profile, project domain, or target style, so it cannot precisely do “expand this, but keep my style.”

**Questions:**

- Since a single TIPO model is used for multiple T2I backbones (rather than training one optimizer per T2I model), does this create subtle distribution-mismatch issues for models whose caption/style format differs from the main training corpus?
- Does the method exhibit a scaling law? In other words, if we move from the 200M model to larger variants, do we see monotonic gains in fidelity, aesthetic score, and OOD robustness — and where is the efficiency/quality turning point?

---

> ### Author Response · Authors · 2025-11-15
>
> Thank you very much for the clear and encouraging review. We sincerely appreciate your recognition of our conceptual framing, the lightweight prefix-preserving design, and the breadth of our experiments. Your comments are very helpful. We now respond to the concerns and questions you raised.
>
> ## Weakness
>
> **Distribution dependent:** We agree that TIPO's input processing is limited by the training corpora. When user prompts are extremely niche or stylistically unusual, the expanded prompts may indeed become less reliable. However, **this limitation applies to all prompt optimizers without strong world knowledge.** For example, GPT-4o-mini may handle such cases by understanding user intent and rewriting unusual prompts into T2I-friendly forms, whereas lighter models (such as the GPT-2 based baselines) struggle significantly. We believe this limitation can be mitigated by initializing TIPO from a stronger LLM backbone rather than training a Llama from scratch. We will clarify this limitation in the revision and highlight it as future work.
>
> **Narrowed generation space and Vendi:** We clarify that Vendi score measures dispersion in the DINOv2 feature space and is sensitive to noise (as stated in its original paper [1]). Intuitively, weakly guided prompts tend to produce more low-level artifacts, which appear as noisy outliers in the feature space and can artificially inflate Vendi without reflecting true semantic diversity. Figure 4 illustrates this clearly: with TIPO, the scenic compositions become visually richer and more varied, yet Vendi decreases because artifacts are substantially reduced (corresponding to Table 3, in-domain tag-based prompts). In this case, a lower Vendi score results from fewer spurious artifacts rather than reduced diversity. We apologize for not explaining this limitation clearly, and will add a short clarification on how to interpret Vendi, which is only meaningful when artifact levels are comparable.
>
> **Closed-source or stylistically distant models:** For closed-source models, we have evaluated TIPO on five leading systems (FLUX.1-dev, Omnigen2, Lumina-2, HiDream-I1, and Gemini-Flash-2.0-Image). As shown in Table 3, four of them show clear aesthetic improvements with TIPO-optimized prompts. This is unsurprising, as TIPO is trained on high-quality open-source text–image corpora (over 30 million prompts with around 40 billion tokens) that are widely used in the community. These prompts are unlikely to be entirely absent from the training data of general closed-source models. For stylistically distant T2I models, we acknowledge that TIPO may introduce slight aesthetic regressions. Such models often employ prompting conventions that differ substantially from mainstream practice, making them harder to use without model-specific adjustments. Yet, we believe our approach remains insightful for the model designers, as it suggests how lightweight, distribution-aware components can be added on their side for more user-friendly prompt interfaces.
>
> **Lack of user-conditioned or style-preserving expansion:** We agree that the current TIPO is a generic optimizer and does not condition on user profile, project domain, or target style. Personalization is an important but separate challenge, and we will highlight it in future work, for example through lightweight adapters or online reinforcement learning to track user-specific preferences.
>
> ## Q1. Distribution mismatch
>
> We agree that using a single TIPO model for multiple T2I backbones can introduce mild distribution mismatch. However, in practice this effect has been limited, largely because our training corpus is both large and diverse, and modern T2I models use strong text encoders that tolerate moderate stylistic variation. In cases of more substantial mismatch (e.g., when a T2I model requires JSON-structured prompts), TIPO can be lightly adapted by finetuning on a small number of model-specific samples or by applying existing RL-based tuning strategies. We will highlight these model-specific adaptations as future work.
>
> ## Q2. Scaling Law
>
> Thank you for the insightful question. We partially investigate scaling in Appendix Table 7 by increasing TIPO from 200M to 500M parameters. Some of the results are reproduced below:
>
> | Metric          | Task     | TIPO-200M | **TIPO-500M** |
> | --------------- | -------- | --------- | ------------- |
> | **FDD ↓**       | NL-short | 0.1529    | **0.1356**    |
> |                 | NL-long  | 0.1650    | **0.1398**    |
> | **Aesthetic ↑** | NL-short | 5.8531    | **5.8943**    |
> |                 | NL-long  | 5.8364    | **5.9030**    |
>
> The 500M model consistently improves over the 200M model on both FDD and Aesthetic. Unfortunately, we were unable to scale beyond 500M to see a turning point in the efficiency–quality tradeoff due to budget constraints. We will note this limitation in the revision.
>
> [1] Dan et al. The Vendi Score: A Diversity Evaluation Metric for Machine Learning. TMLR 2023

---

> > ### Author Response · Authors · 2025-11-22
> > **Manuscript Update**
> >
> > Thank you again for your thorough review. Our main revisions are:
> >
> > - Added a clarification in Section 5.1 metrics part explaining that Vendi scores reflect feature‑space noise and should accompany quality metrics.
> > - Expanded Appendix J to discuss distribution dependence and OOD generalization, including suggestions for stronger backbones, domain‑diverse fine‑tuning and larger user prompt collections.
> > - Added future directions into Appendix J on stronger backbone initialization, model‑specific adaptation, personalization and scaling behaviour.
> >
> > For your convenience, all edits in the manuscript are marked in **blue** in the revised PDF.
> > We hope these updates address your concerns.

---

### Official Review · Reviewer_fBC3 · 2025-11-01

**Soundness:** 4
**Presentation:** 3
**Contribution:** 3
**Rating:** 8
**Confidence:** 4

**Summary:**

In this paper, the authors propose a new method for prompt rewriting for text-to-image generation. They design a specific format to structurally combine natural language and tag-based prompts and implement a pre-sampling algorithm that progressively refines arbitrary, coarse user input into organized, fine-grained prompts by training LLMs. They evaluate their models against prior baselines as well as SOTA LLMs on a large variety of text-to-image models as well as native multimodal models using standardized metrics and human preference survey.

**Strengths:**

1. This paper has very comprehensive experiments, especially the range of text-to-image models and native multimodal models that they test with. And the fact that it is effective even for models with self-refinement capabilities also shows good practical applicability of their method.
2. ELO rating is an interesting and creative way to showcase human evaluation comparison results among multiple models. I don’t think this is a standard metric for image generation evaluation yet, but I think it should become one.
3. The description of their training recipe is also very clear and easy to follow, making the method also very adoptable.
4. The inference speed is very fast.
5. The models are trained with relatively small GPUs in a relatively short amount of time, which makes the method resource efficient.

**Weaknesses:**

1. The improvement on benchmarks that TIPO brings is not very consistent across the board and is sometimes very marginal.
2. Mild writing suggestions: Section 3 does not really provide much information and is not very well connected to the rest of the paper. Given how packed the remainder of the paper is, I would suggest either shortening that section or removing it entirely and just explaining the notations when using them later on.
3. Minor: there are some inconsistencies in citation styles in the first paragraph of the introduction.

**Questions:**

From the qualitative examples, it seems like the diversity of the images should be improved after rewriting. However, this is not reflected with the Vendi scores. Do the authors have an explanation of why this may be the case?

---

> ### Author Response · Authors · 2025-11-15
>
> Thank you very much for your thoughtful and encouraging review. We sincerely appreciate your recognition of the strengths of our work, such as the comprehensive experimental coverage, the practical applicability across diverse T2I models, and the creative use of ELO for human evaluation. Your positive feedback is greatly motivating for us.
>
> We also appreciate the constructive points you raised, and we address them below.
>
> ## W1. Consistency of benchmark improvements
>
> We acknowledge that the improvements are not uniform across every metric and dataset. This is partly because the original input prompts and each baseline has their own strengths. As shown in Table 2 of our paper, for example:
>
> - Raw input prompts are the shortest and contain minimal information, so the T2I model receives the weakest guidance. Since all our T2I models rely on random sampling, this naturally leads to the highest Vendi score but very low aesthetic scores.
> - GPT rewritten prompts contain the richest world knowledge and often resemble expert artistic guidance, producing highly coherent and elegant images and therefore the highest aesthetic scores, but at the cost of reduced fidelity and diversity.
>
> As shown above, these metrics are inherently in tension with one another, and the baselines are particularly strong in specific aspects, making consistent improvement across all metrics difficult. TIPO instead focuses on aligning with large scale T2I training text corpora, which reduces artifacts (i.e., the lowest AI Corrupt scores) while keeping all other metrics competitive. This balanced strategy is empirically successful, reflected in our highest average ranking across quantitative metrics and the strongest preference in user evaluations.
>
> ## W2. Writing of Section 3
>
> Thank you for the precise writing suggestion. We fully agree that Section 3 was not well connected to the rest of the paper. We will remove the standalone notation part and introduce notation only when it is first needed. At the same time, we will keep a minimal amount of basic model and problem definitions to maintain clarity for readers who are less familiar with the setting. This restructuring indeed makes the logical flow much smoother.
>
> ## W3. Citation styles
>
> Thank you for the detailed proofreading. We will correct the inconsistencies immediately.
>
> ## Q. On the discrepancy between qualitative diversity and Vendi scores
>
> Thank you for raising this question. A higher Vendi score does not necessarily indicate more meaningful visual diversity, because Vendi measures dispersion in the DINOv2 feature space rather than semantic variation. Because Vendi is defined directly on a similarity matrix in feature space [1], any non-semantic variations captured by the encoder (including artifacts) can affect the score. In our setup with DINOv2 features, this means that noise-like artifacts may inflate Vendi even when semantic diversity does not increase. When the original prompts are very short, the T2I model receives weak guidance and tends to generate more low-level artifacts, as also reflected by the AI Corrupt Score. These artifacts may introduce irregular noise patterns in the feature space and can unintentionally inflate Vendi without corresponding semantic diversity. In practice, Vendi becomes meaningful only when artifact levels are comparable across methods. TIPO achieves competitive diversity while producing far fewer artifacts, which aligns with the qualitative differences shown in Figure 4.
>
> [1] Dan et al. The Vendi Score: A Diversity Evaluation Metric for Machine Learning. TMLR 2023

---

> > ### Author Response · Authors · 2025-11-22
> > **Manuscript Update**
> >
> > Thank you again for your helpful feedback. We have uploaded a revised manuscript addressing your suggestions:
> >
> > - **Re‑wrote Section 3**: The notation definitions are removed from the Preliminaries section for better flow.
> > - **Citation cleanup**: We corrected inconsistent citation styles throughout the paper.
> > - **Vendi clarification**: Section 5.1 metrics part now notes that Vendi measures feature‑space dispersion and should be used with quality metrics.
> >
> > For your convenience, all edits in the manuscript are marked in **blue** in the revised PDF.
> > We appreciate your review and hope the revisions improve clarity.

---

> > > ### Comment · Reviewer_fBC3 · 2025-11-25
> > >
> > > Thank you for your rebuttal. It completely resolved all my questions and concerns and I will maintain my score of 8.

---

> > > > ### Author Response · Authors · 2025-11-26
> > > >
> > > > We thank Reviewer fBC3 for the invaluable feedback on our response.

---

### Author Response · Authors · 2025-12-02
**General Response**

We sincerely thank the reviewers for their constructive feedback and the time invested in evaluating our work.

We are greatly encouraged that the reviewers value TIPO as a clear, universally compatible, and efficient framework. By **aligning user inputs with the training text distributions of T2I models**, our approach proves effective across a wide spectrum of architectures while achieving better scalability than heavy reinforcement learning pipelines.

In particular, the reviews highlighted the following strengths:

* **Conceptual Clarity:** Reviewers praised our core intuition of "distribution alignment" as novel and straightforward. They noted that treating prompt optimization as a problem of bridging the gap between user input and training data provides an interesting insight (highlighted by Reviewers TMB7, LKuZ, TZN3).
* **Compatibility:** Our framework was commended for its effectiveness across diverse generative models. Its ability to improve results for both open-source models and proprietary systems (such as FLUX and commercial APIs) demonstrates strong practical applicability (highlighted by Reviewers fBC3, LKuZ, TZN3).
* **Scalability:** The reviewers recognized the method's computational efficiency. Unlike resource-intensive LLM rewriting or RL-based approaches, TIPO offers a lightweight solution that is easy to adopt, train, and deploy (noted by all reviewers).
* **Evaluation:** The comprehensive experimental validation was highlighted as a key strength (noted by all reviewers). Specifically, the adoption of ELO ratings for human preference was noted as a creative and interesting way to benchmark performance (highlighted by Reviewer fBC3).

To address the reviewer’s concerns, we have strengthened the manuscript with new experiments and analyses:

* **Direct Validation of Alignment:** We incorporated direct embedding distance measurements and CLIPScore evaluations, which empirically verified our distribution alignment hypothesis. *(Suggested by Reviewer LKuZ in W1/Q1 and W4/Q4; added Table 5 and 6 in Section 5.1)*
* **Training Cost Comparison:** We reported the direct comparison of training costs highlighting TIPO's advantages over RL-based baselines. *(Suggested by Reviewer LKuZ in W3/Q3; added Table 13 in Appendix H.5)*
* **Clarifying Vendi:** We improved the explanation of the Vendi score to clarify that it reflects feature-space dispersion and should accompany quality metrics. *(Addressing Reviewer fBC3 and TMB7; updated Section 5.1)*
* **Future Directions:** We have consolidated discussions on limitations, including out-of-distribution generalization and potential image-feedback mechanisms, into the revised Appendix. These sections outline clear paths for future research while maintaining the focus on our current contributions. *(Addressing Reviewers TMB7, LKuZ W2/Q2, TZN3; added to Appendix J)*
* **Sampling Strategy:** We clarified sampling hyperparameters used in inference. *(Addressing Reviewer TZN3; added to Appendix C.3)*
* **Writing Improvements:** We streamlined Section 3 and unified citation styles. *(Addressing Reviewer fBC3 W2–W3)*

All revisions in the paper are marked in *blue*.

Once again, we thank all reviewers for their constructive feedback. It is deeply regrettable that the unexpected OpenReview incident led to an early score being reverted (LKuZ 6→4) and made further direct discussions with reviewers impossible. We also understand the additional workload this has caused for the area chair. However, we sincerely hope to receive further comments through the meta-review process, as the quality of our paper has significantly improved following reviewers’ suggestions (highlighted by Reviewer LKuZ). We remain fully committed to refining the manuscript based on your valuable feedback, so as to ensure it is at its best.

---

### Meta-Review · Area_Chair_t7DM · 2026-01-06

**Summary:**

Most reviewers have positive initial reviews for the paper (initial score 8, 6,6,4).

The major concerns from the reviewers are mainly about  the proposed method (which is distribution-dependent and needs large-scale open-sourced training data), and the experiment (e.g., the improvement on benchmarks is inconsistent, the verification of "distribution-aligned" is indirect, no comparison on training cost, lack of text-image alignment metrics).  After rebuttal, most concerns are resolved, including those from reviewer LKuZ whose initial score is 4.

**Reviewer Concerns:**

Most concerns are resolved. The major concern remains seems to be the fact that the proposed method is distribution-dependent and needs large-scale open-sourced training data, and hence may not apply to emerging models trained on proprietary datasets. However, as quoted from reviewer LKuZ:  "it does not undermine the core value of the proposed method".

**Reviewer Scores:**

reviewer LKuZ might increase the score from 4 to 6.

---

### Decision · Program_Chairs · 2026-01-26

Accept (Poster)